

**Aerosol optical properties derived from POLDER-3/PARASOL (2005-2013) over the western Mediterranean Sea: I. Quality assessment with AERONET and in situ airborne observations**

Paola Formenti[1,*], Lydie Mbemba Kabuiku[1,4], Isabelle Chiapello[2], Fabrice Ducos[2], François Dulac[3] and Didier Tanré[2]

[1] Laboratoire Interuniversitaire des Systèmes Atmosphériques, UMR CNRS 7583, Université Paris-Est Créteil et Université Paris Diderot, Institut Pierre-Simon Laplace, Créteil, France

[2] Laboratoire d'Optique Atmosphérique, UMR CNRS 8518, Université Lille, Villeneuve d'Ascq, France

[3] Laboratoire des Sciences du Climat et de l'Environnement, UMR 8212 CEA-CNRS-UVSQ 8212, Institut Pierre-Simon Laplace, Université Paris-Saclay, Gif-sur-Yvette, France

[4] Agence De l'Environnement et de la Maîtrise de l'Energie (ADEME) 20 avenue du Grésillé,- Angers, France

* corresponding author (paola.formenti@lisa.u-pec.fr)

For submission to Atmos. Meas. Tech., ChArMEx special issue



**Abstract**
The western Mediterranean atmosphere is impacted by a variety of aerosol sources, producing a
complex and variable mixture of natural and anthropogenic particles, with different chemical and
physical properties. Satellite sensors provide a useful global coverage of aerosol parameters but
through indirect measurements that request careful validation. Here we present the results of a long-
term regional scale analysis of the full dataset (March 2005 and October 2013) of POLDER-
3/PARASOL ocean operational retrievals of the total, fine and coarse aerosol optical depth (AOD,
$AOD_F$ and $AOD_C$), Angstrom exponent (AE), and the spherical/non-spherical partition of coarse-mode
AOD ($AOD_{CS}$ and $AOD_{CNS}$), respectively. The evaluation is performed using data from seventeen
coastal and insular ground-based AERONET sites on one side, and airborne vertical profiles of
aerosol extinction and number size distribution obtained by the SAFIRE ATR 42 aircraft operated in
the area during summer 2012 and 2013 on the other side. This study provides the first regional
evaluation of uncertainties of the POLDER-3 products, and highlights their quality. The POLDER-3
Ångström exponent, representing AOD spectral dependence in link with the aerosol particle size
distribution, is biased towards small values. This bias, however, does not prevent using AE for
classifying the regional aerosol laden air masses. $AOD_F$ corresponds to particle smaller than 0.6-0.8
µm in diameter and appears suitable to monitor the aerosol submicron fraction from space. We also
provide an original validation of POLDER-3 $AOD_C$ and its spherical/non-spherical partition, which
shows agreement within 25% with AERONET shape retrievals when the aerosol coarse fraction
dominates.
**1. Introduction**
Aerosols include a large variety of particles (mineral dust, sea salt, soot carbon and organic species,
sulphates, nitrates…) emitted by natural and anthropic sources and different mechanisms
(combustion, wind erosion, gas-to-particle conversion, etc.). Aerosols have a short lifetime in the
troposphere (Boucher, 2015) but they are key to many atmospheric processes, as the redistribution
of solar and thermal radiation by scattering and absorption, cloud formation and precipitation, and air



quality degradation, which, in turn are relevant in shaping the Earth climate and liveability (Pope III et
al., 2002; Akimoto, 2003; Pope III and Dockery, 2006; Monks et al., 2009; Boucher et al., 2013).
Despite its importance, the global aerosol radiative effect is far from being certain, as both aerosol
spatial distribution and optical properties are affected by large unknowns (Boucher et al., 2013; Myhre
et al., 2013). Furthermore, the apportionment of aerosols to anthropic and natural sources is critical
to evaluate the perturbative forcing of human activities on the Earth radiative budget and ultimately
climate (Myhre et al., 2013; Shindell et al., 2013; Kim et al., 2014; Pan et al., 2015). In this general
context, the Mediterranean basin is a region of great interest. Submitted to demographic pressure
and experiencing bad air quality (Monks et al., 2009; Kovats et al., 2014), the Mediterranean is a high
emission and transport region of all kinds of anthropogenic and natural aerosols (e.g. Moulin et al.,
1998; Lelieveld et al., 2002; Pace et al., 2005 and 2006; Querol et al., 2009; Pey et al., 2013; Becagli
et al., 2017), as well as one of the most vulnerable areas to climate change (Giorgi, 2006), with severe
future warming leading to a reduction in precipitations and soil moisture, and henceforth a significant
water stress towards the end of the century (Giorgi and Lionello, 2008; García-Ruiz et al., 2011;
Christensen et al., 2013).
Through the years, the Mediterranean aerosols have been investigated through a number of
dedicated local and regional scale experiments (e.g. Söderman and Dulac, 1998; Formenti et al.,
2002; Lelieveld et al., 2002; Zerefos et al., 2002; Dulac and Chazette, 2003; Cros et al., 2004; Putaud
et al., 2004, Mallet et al., 2016), surface monitoring stations and networks (e.g. Bergametti et al.,
1989; Migon et al., 1993; Mihalopoulos et al., 1997; Meloni et al., 2007; di Sarra et al., 2008; Pérez
et al., 2008; Querol et al., 2009; Kalivitis et al., 2011; Mallet et al., 2013; Pappalardo et al., 2014;
Lyamani et al., 2015) and satellite observations (e.g. Dulac et al., 1992; Moulin et al., 1998; Barnaba
and Gobbi, 2004; Antoine and Nobileau, 2006; Papadimas et al., 2008; Gkikas et al., 2009 and 2016).
More recently, the regional-scale Chemistry-Aerosol Mediterranean Experiment (ChArMEx,
http://charmex.lsce.ipsl.fr/) within the international Mediterranean Integrated STudies at Regional And
Local Scales (MISTRALS, http://www.mistrals-home.org) program has significantly added to the





existing body of knowledge by providing new ground-based, airborne and balloon-borne observations
over the western part of the basin (Mallet et al., 2016; see also this special issue).
ChArMEx has also provided a new momentum in the analysis of regional ground-based and satellite
aerosol observations on long and short periods (e.g. Mallet et al., 2013; Nabat et al., 2013; Lyamani
et al., 2015; Gkikas et al., 2016; Granados-Muñoz et al., 2016; Sicard et al., 2016). Satellite data are
highly valuable to provide information on the regional and global aerosol spatial and temporal
distribution and optical properties which are input to climate models. Most satellite instruments (e.g.,
MODIS, SEAWIFS, AVHHR, SEVIRI…) retrieve the Aerosol Optical Depth (AOD), representing the
column-integrated optically-active content of atmospheric aerosols, and also proportional to the net
change in the clear sky outgoing radiative flux at the top of the atmosphere (Boucher, 2015). The
AOD is an essential parameter to establish the climatology of the distribution and effects of
atmospheric aerosols and it is often used for model evaluation (Nabat et al., 2013). With this respect,
advanced spaceborne retrievals deriving the AOD as a function of particle size and shape, and
possibly of wavelength, are most useful in evaluating the origin and the radiative effect of aerosols of
different nature.
In this paper, we present a first comprehensive quality-assessment study of the advanced dataset
provided by the operational retrieval ocean algorithm of the third multi-spectral, multi-directional and
polarized POLDER-3 (POLarization and Directionality of the Earth's Reflectances) radiometer on
PARASOL (Polarization & Anisotropy of Reflectances for Atmospheric Sciences coupled with
Observations from a Lidar) satellite (Herman et al., 2005; Tanré et al., 2011) over the western
Mediterranean basin. POLDER-3 operated from March 2005 to October 2013 and provided the total,
fine and coarse mode aerosol optical depth (AOD, $AOD_F$ and $AOD_C$) at the wavelength of 865 nm,
the spectral dependence of the AOD (Angström Exponent, AE), and the partition of spherical and
non-spherical $AOD_C$ ($AOD_{CS}$ and $AOD_{CNS}$, respectively). This paper extends previous evaluations of
AOD and $AOD_F$ (Goloub et al., 1999; Fan et al., 2008; Bréon et al., 2011), and provides the first
estimate of the significance of the coarse mode spherical and non-spherical components ($AOD_C$,
$AOD_{CS}$ and $AOD_{CNS}$).





This study is based on comparisons with co-localised observations from the sun/sky photometers of
coastal and insular stations of the Aerosol Robotic Network (AERONET; Holben et al., 1998), and
with the in situ measurements of vertical profiles of aerosol extinction and size distribution which were
performed by the French ATR 42 environmental research aircraft of the Service des Avions Français
Instrumentés pour la Recherche en Environnement (Safire, www.safire.fr) during the ChArMEx
intensive campaigns (Di Biagio et al., 2016, Denjean et al., 2016, Mallet et al., 2016). In particular,
the use of the size distribution vertical profiles measured in situ allows us to calculate the aerosol
optical depth over different size ranges, and the evaluation of $AOD_F$ and $AOD_C$.
The analysis presented in this paper is essential to geophysical analyses of observations by
POLDER-3 of the spatial and temporal variability of the aerosol load over the western Mediterranean
basin.
**2. Measurements**
**2.1. POLDER-3/PARASOL**
The third radiometer POLDER-3 on PARASOL, operational from March 2005 to October 2013, was
part of the A-Train constellation operated on a sun-synchronous orbit at 705 km crossing the Equator
at 13:30 (Equator local time) (Tanré et al., 2011). In December 2009, it left the A-Train, and continued
the observations at 3.9 km below, and at 9.5 km below in 2011. This changed its hour of passage,
which was 16:00 Equator local time at the end of the operational period.
POLDER-3/PARASOL used a 274 x 242-pixels CCD detector array, each pixel covering 5.3 x 6.2 km²
at nadir. The size of the POLDER-3 images was 2100 x 1600 km², allowing to achieve a global
coverage within two days. The western Mediterranean area could be covered in less than 5 minutes
along its north-to-south axis. The spatial resolution of POLDER-derived (Level 2) aerosol parameters
is about 18.5 x 18.5 km² (corresponding to 3 x 3 pixels of the Level-1 grid; http://www.icare.univ-
lille1.fr/parasol/products).
The instrument measured solar radiance at 9 wavelengths from 443 to 1020 nm, three of which with
polarisation (490, 670, 865 nm), and at up to 16 different angles (±51° along, ±43° across track).





Cloud screening according to Bréon and Colzy (1999) was applied to minimize possible cloud
contamination of aerosol products.
In this paper, we used the latest algorithm update (collection 3) performed in 2014 of the operational
clear-sky ocean retrieval algorithm (Deuzé et al., 1999, 2000; Herman et al., 2005). This latest version
includes calibration improvements and uses the total and polarized radiances at 670 and 865 nm. For
each clear sky pixel, the algorithm recalculates the observed polarized radiances at several
observational angles from a Look-Up Table (LUT) built on aerosol micro-physical models. These are
constructed as follows: (i) aerosol are not-absorbing, that is, the imaginary part $m_i$ of their complex
refractive index ($m = m_r$ -i $m_i$) is nul. Only the real part $m_r$ is attributed, and considered as invariant
with wavelength between 670 and 865 nm; (ii) the aerosol number size distribution is bimodal and
lognormal with a fine mode with effective diameter ($D_{eff}$) smaller than 1.0 µm and a coarse mode with
$D_{eff}$ larger than 1.0 µm. The coarse mode includes a non-spherical fraction based on the spheroidal
model from Dubovik et al. (2006). Collection 3 increases the number of modes with respect to the
previous versions reported by Herman et al. (2005) and Tanré et al. (2011), and allows spheroidal
$D_{eff}$ to take two values (2.96 or 4.92 µm). The summary of LUT parameters are presented in the
supplementary material (**Table S1**).
The calculations of the multi-spectral, multi-angle polarized radiances are done using a Mie model for
homogeneous spherical particles or the spheroidal optical model developed by Dubovik et al. (2006).
A quality flag index (0 indicating the lowest and 1 the highest quality) is attributed to each pixel
depending on the quality of radiance simulation.
In this paper, we target the following POLDER-3 oceanic (i.e. over ocean surfaces) aerosol products,
in which AODs are at 865 nm:
•      The total aerosol optical depth (AOD),and the Ångström Exponent (AE) representing the

spectral dependence of AOD, and calculated as


$$AE = - \frac{\ln(AOD_{865}/AOD_{670})}{\ln(865/670)} \tag{1}$$




•  The aerosol optical depth due to the fine particle mode ($AOD_F$)
•  And the aerosol optical depth due to the spherical ($AOD_{CS}$) and non-spherical ($AOD_{CNS}$) coarse
mode fractions, obtained for clear-sky pixels with favourable viewing geometries (scattering
angles between 90° and 160°). These products allow estimating the fraction of non-spherical
particles in the coarse mode AOD ($f_{CNS}$) from

$$f_{CNS} = AOD_{CNS} / (AOD_{CNS} + AOD_{CS}) \qquad (2)$$

Whereas $AOD_F$ was available for all clear-sky pixels regardless of the geometry of observations, the
$AOD_C$ was estimated in two ways depending on the availability of observations. For days with
observations in favourable viewing geometrical conditions, $AOD_C$ was calculated as the sum of
measured $AOD_{CS}$ and $AOD_{CNS}$. For the remaining days, $AOD_C$ was calculated as $AOD–AOD_F$. A
maximum difference of ±0.002 rounding errors was found for days when both methods are applicable.
Only the POLDER-3 aerosol products from pixels with a quality flag index ≥0.5 have been considered
in the following discussion.
**2.2. AERONET**
AERONET is a global network of ground-based multi-spectral sun/sky photometers (Holben et al.,
1998; 2001) dedicated to real time monitoring of aerosol properties and widely used as ground-based
reference for validation of aerosol satellite retrievals (e.g., Goloub et al., 1999; Bréon et al., 2011). It
uses standardized sun/sky photometers (CIMEL CE-318, Cimel Electronique, Paris) measuring solar
extinction and sky radiances (at times with polarization) in the almucantar plane at wavelengths
between 340 and 1020 nm (most commonly 440, 675, 870, and 1020 nm), that allow deriving a
number of aerosol optical and microphysical parameters (Dubovik and King, 2000; Dubovik et al.,

2006).

AOD and AE are obtained about every 15 minutes from the measurement of the direct sun extinction
and are reported as the average of a triplet of acquisitions lasting approximately 30 s. For freshly
calibrated and well maintained instruments, the accuracy in AOD is of the order of 0.01-0.02



regardless of the AOD value (Holben et al., 1998). The aerosol optical depth in the fine and coarse
mode ($AOD_F$ and $AOD_C$, respectively) are recalculated from the column-integrated volume size
distribution retrieved by the inversion algorithm described in Dubovik and King (2000) and Dubovik et
al. (2006). The fine and coarse modes of the retrieved volume size distribution are defined as the
modes below and above a threshold diameter ($D_{cut-off}$) corresponding to the minimum of the size
distribution. The $D_{cut-off}$ value can vary between 0.44 and 0.99 µm. $AOD_F$ and $AOD_C$ values are
estimated by recalculating the extinction due to the fine and coarse modes of the aerosols. The latest
AERONET retrieval scheme considers an aerosol mixture of polydisperse, randomly-oriented
homogeneous spheroids with a fixed distribution of aspect ratios (Mishchenko et al., 1997) and
provides fraction (in percentage) of non-spherical/spherical particles, i.e. $f_{NS}/f_S$ (Dubovik et al., 2006).
By clear sky, there are about 10 measurements per day of this fraction in the early day or late
afternoon (solar zenith angle ≥50°).
We used AERONET V2 level-2 quality assured aerosol products. Seventeen coastal AERONET
stations, shown in **Figure 1,** were selected in this study, (see also **Table 1** for their respective
geographical coordinates and covered periods). Their regional distribution covers the entire western
Mediterranean basin, including south Europe (e.g., near coastal stations of Barcelona, Toulon,
Villefranche-sur-Mer…), North Africa (Blida), and island locations in the northern (Ersa), central
(Palma de Mallorca) and southern (Lampedusa and Alboran) basin, therefore capturing the diversity
of the aerosol population, resulting from the different sources contributing to the Mediterranean
aerosol (desert dust, marine, urban and industrial pollution, and biomass burning). The dataset also
includes the ground-based super-sites of Ersa and Lampedusa of the ChArMEx project (Mallet et al.,
2016). Considering the 17 stations altogether, more than 18000 daily observations of AOD are
available in total in both POLDER-3 and AERONET datasets, among which 6421 are concurrent (see
section 3.2 below) and thus available for comparison. We did not consider for tentative matching with
POLDER in this study a rather limited number (<100) of daily observations obtained from manual sun
photometers on-board ships in our area (Figure 1) and period of interest, which are also available
from the Maritime Aerosol Network component of AERONET (Smirnov et al., 2011).





**2.3.    ChArMEx airborne measurements**

The airborne measurements relevant to this paper were performed on the French ATR 42 environmental research aircraft of Safire during two of the intensive observational periods of the ChArMEx project:

- The Transport and Air Quality (TRAQA) campaign, dedicated to the study of air pollutants transport from Europe to the Mediterranean, their evolution and their impact on regional air quality (Di Biagio et al., 2015; 2016; Nabat et al., 2015a; Rea et al., 2015);

- The Aerosol Direct Radiative Forcing on the Mediterranean (ADRIMED) campaign was dedicated to the characterization of aerosol optical properties in the Mediterranean and their direct radiative effect in clear sky conditions (Denjean et al., 2016; Mallet et al., 2016).

During TRAQA, the ATR 42, based at the Francazal airport near Toulouse, France (43°36'N, 1°26'E), conducted 17 flights from 20 June to 13 July 2012 encountering weather conditions favouring the transport of pollution aerosols from continental Europe, and particularly from the Rhone valley, the Gulf of Genoa and Barcelona, giving raise to AOD values in the range of 0.2-0.6 at 550 nm over the northwestern Mediterranean. From 17 to 23 June, and then on 29 June, two episodes of desert dust transport were observed in the free troposphere, increasing the AOD up to 1.4 on June 29. (Di Biagio et al., 2015; 2016). During ADRIMED, the ATR 42, based in Cagliari, Italy (39°15'N, 9°03'E), flew 16 scientific flights between 14 June and 4 July 2013 (Denjean et al., 2016; Mallet et al., 2016). Several episodes of desert dust transport from southern Algeria and Morocco and northern Algeria and Tunisia were observed over the western and central Mediterranean, particularly off the Balearic Islands and above the Lampedusa island offshore Tunisia (Denjean et al., 2016).The total optical depth at 550 nm remained moderate, in the order of 0.2-0.4 even during dust events (Mallet et al., 2016).

**2.3.1.  Airborne instrumentation measuring aerosol optical properties**

**2.3.1.1.      PLASMA photometer**

PLASMA (Photomètre Léger Aéroporté pour la Surveillance des Masses d'Air), developed by LOA (Laboratoire d'Optique Atmospherique, Lille), is a multi-spectral sunphotometer which measures the



direct sun radiance and retrieves the AOD at 15 wavelengths between 343 and 2250 nm, including
869 nm (Karol et al., 2013). The estimated uncertainty ranges between 0.005 and 0.01 (Karol et al.,
2013). PLASMA was operated during the ADRIMED campaign only, when it was mounted on the roof
of the ATR 42, allowing the retrieval of a vertical profile of both the spectral AOD and the aerosol
particle size distribution (Torres et al., 2017).
**2.3.1.2.        CAPS-PMex**
The Cavity Attenuated Phase Shift in situ instrument (CAPS-PMex, Aerodyne Research Inc.)
measures the extinction coefficient $\sigma_{ext}$ at 532 nm with an estimated relative uncertainty of ±3.2%
(Kebabian et al., 2007; Massoli et al., 2010; Petzold et al., 2013). The operating principle is based on
the modulation of the frequency and the phase changes of the light emitted by a LED source due to
aerosols, after correction of the Rayleigh scattering by the molecules present in the air mass. As
described in Denjean et al. (2016), the instrument was available during the ADRIMED campaign only,
when it was located inside the cabin behind the Communautary Aerosol Inlet (CAI), and operated at
0.85 L min$^{-1}$ and with a temporal resolution of 1 second. In this paper, the extinction coefficient $\sigma_{ext}$ is
expressed in Mm$^{-1}$ (1 Mm$^{-1}$ = 10$^{-6}$ m$^{-1}$).
**2.3.1.3.        Nephelometer**
The scattering coefficient $\sigma_{scatt}$ at 450, 550 and 700 nm was measured by a spectral integrating
nephelometer (model 3563, TSI Inc.) described extensively by Anderson et al. (1996) and Anderson
and Ogren (1998). During both TRAQA and ADRIMED, the instrument was operated at 30 L min$^{-1}$
with a temporal resolution of 1-2 seconds downstream the AVIRAD inlet also onboard the ATR 42 (Di
Biagio et al., 2015; 2016; Denjean et al., 2016). The AVIRAD inlet estimated size cut-off,
corresponding to the diameter at which particles are collected with a 50% efficiency, is 12 μm in optical
diameter.
The instrument uses a halogen lamp as light source and three photomultipliers preceded by spectral
filters. Due to the geometry of its sensing volume, the nephelometer measures the scattering
coefficient ($\sigma_{scatt}$) between 7° and 170° and the backscattering coefficient ($\sigma_{bscatt}$) between 90° and



170°. The scattering Angström exponent AE$_{scatt}$ and representing the scattering spectral dependence
car be calculated as

$$AE_{scatt} = -\frac{\ln(\sigma_{scatt,450}/\sigma_{scatt,700})}{\ln(450/700)}$$
(3)


The relative uncertainty in σ$_{scatt}$ due to calibration, counting statistics and non-idealities of detector
surfaces, is estimated to be ±1-2% for submicron aerosols and ±8-15% for supermicron aerosols
(Müller et al., 2009). To these values usually adds the error related to the geometric truncation of the
measured angular range of the scattering phase function due to the sensing volume (Anderson and
Ogren, 1998). This truncation induces an underestimation of σ$_{scatt}$ and σ$_{bscatt}$, which depends on the
angular distribution of the scattered light, and thus on particle size. Anderson and Ogren (1998) have
shown that the uncertainty induced by the underestimation of σ$_{scatt}$ can be parameterized by the
scattering spectral dependence for submicron aerosols. This parameterization is not possible for
aerosols of larger size (diameter greater than 1 μm), because the Angström coefficient tends to zero
whereas the underestimation is important (50-60%) because of the increase of the forward scattering.
In this case, the correction is performed by optical calculation if the particle size distribution and
refractive index are known (Müller et al., 2009; Formenti et al., 2011). As for σ$_{ext}$, in this paper σ$_{scatt}$ is
expressed in Mm$^{-1}$.
**2.3.2. Aerosol particle size distribution**
Because of its extent, the aerosol particle size distribution is measured in situ by the combination of
several instruments, often based on different physical principles (Wendisch and Brenguier, 2013). In
our work, we used a combination of different optical counters operating on the fine and coarse modes
of the aerosols, that is:
• a Passive Cavity Aerosol Spectrometer Probe (PCASP, Droplet Measurement Technologies,
Boulder, Colorado), operated at 632.8 nm with a temporal resolution of 1 second. The PCASP





measures light scattering between 35 and 135° to derive the particle number size distribution

over 31 channels between 0.1 and 3.0 µm in diameter (Liu et al., 1992; Reid et al., 1999). The

PCASP was operated on a wing pod of the ATR 42 during the TRAQA campaign only.

• an Ultra High Sensitivity Aerosol Spectrometer (UHSAS, Droplet Measurement Technologies,

Boulder, Colorado), operated at 1054 nm with a temporal resolution of 1 second. The UHSAS

measures light scattering between 22 and 158° to derive the particle number size distribution

over 99 size channels between 0.04 and 1.0 µm in diameter (Cai et al., 2008). The UHSAS

replaced the PCASP under the aircraft wing during the ADRIMED campaign.

• a Sky-Grimm counter (1.129 model, Grimm Aerosol Technik; Grimm and Eatough, 2009),

operated at 632.8 nm with a temporal resolution of 6 seconds. The instrument integrates light

scattering between 30° and 150° to derive the particle number size distribution over 32

channels between 0.25 and 30µm in diameter (Grimm and Eatough, 2009). The instrument

was available during both TRAQA and ADRIMED, operated inside the aircraft cabin and

behind the AVIRAD inlet. Due to a flow problem, measurements during TRAQA are restricted

to the portions of the flights when the ATR 42 remained below 350 m above sea level.

**3. Validation strategy**
**3.1.    Matching POLDER-3 and in situ aircraft measurements**
In situ aircraft measurements provided direct and indirect observations for validation. Direct
observations of the total AOD were obtained by the reading of the PLASMA sun photometer for those
portions of the flights when the ATR 42 flew at its lowest altitude and by integrating the vertical profile
of the extinction coefficient $\sigma_{ext}$ measured by the CAPS-PMex instrument between the minimum and
the maximum heights ($z_{min}$ and $z_{max}$) of the ATR 42 during profile ascents or descents.
Indirect validation of the size-dependent optical depth (AOD, $AOD_F$ and $AOD_C$) was performed by
optical calculation from the number size distribution dN(D,z)/dlogD measured by the combination of
the PCASP, UHSAS and Grimm optical counters as



$$AOD_x \ (865 \ nm) = \int_{z_{min}}^{z_{max}} dz \ \sigma_{ext}(z) = \int_{z_{min}}^{z_{max}} dz \int_{D_{x'}}^{D_x} \pi D^2 Q_{ext}(z, D, m) \frac{dN(D,z)}{dlogD} dlogD \quad (4)$$


The suffix x in Equation 4 indicates the size domain of the aerosol optical depth (total, fine or coarse)
considered in the calculations.
Equation 4 allows one to estimate the aerosol optical depth over a variable size domain, whose
boundaries ($D_{min'}$ and $D_{max}$) can be adjusted to represent the fine and the coarse modes, as well as
the total particle size distribution.
The iterative procedure used for the calculation is presented in **Figure 2**. All calculations used the
optical Mie theory for homogeneous spherical particles (Mie, 1908). The initial step of the procedure
consisted in estimating the aerosol number size distribution, input of Equation 4, from the
measurements of the PCASP, UHSAS and Grimm optical counters operated on board the ATR 42
during TRAQA and ADRIMED. This required two actions, described in details in the Supplementary
material**.**
1.   The conversion of the nominal "optical equivalent spherical diameter" ($D_{EO}$) characteristic of

each particle counter to a "geometric equivalent spherical diameter" ($D_{EG}$). The operating

principle of the particle optical counters is based on the angular dependence of the light

scattering intensity to the particle size (Wendisch and Brenguier, 2013). The proportionality

factor between angular light scattering and particle size depends on the particle complex

refractive index. At calibration, the optical particle counters provide with "an optical equivalent

spherical diameter" ($D_{EO}$), corresponding to the diameter of standard material, generally

spherical latex beeds, which refractive index ($m_{latex}$ = 1.59-0i) is usually different from the real

aerosol refractive index measured in atmosphere. It is therefore necessary to convert the

measured $D_{EO}$ value into a so-called "geometric equivalent spherical diameter" ($D_{EG}$) value

taking into account the actual refractive index of ambient particles.

2.   The combination of measurements over different size ranges. Since no optical counter

completely covers the full size range of atmospheric aerosols, measurements of the PCASP,



UHSAS and Grimm were combined by examining their agreement on their size overlap
domains. When successful, the particle number size distribution obtained by the combination
was normalised to the total particle number and fitted using a multi-mode lognormal
distribution to eliminate discontinuities and extend the representation beyond the lower and
upper operating size ranges of the optical counters.
The capability of the derived number size distributions to represent the aerosol extinction coefficient,
henceforth to estimate aerosol optical depth, was assessed by comparing the calculated extinction
and scattering coefficients $\sigma_{ext}$ and $\sigma_{scatt}$ to the measurements of the CAPS-PMex and the
nephelometer at 450, 532, 550 and 700 nm. The scattering coefficient $\sigma_{scatt}$ was calculated by
integrating the scattering phase function between 7° and 170°, corresponding to the aperture of the
sensing volume of the nephelometer.
All optical calculations performed in this paper assumed the spectral complex refractive index m,
representing the aerosol composition, as independent of size. An initial dataset per aerosol type was
chosen (Table S2 in the Supplementary material). The calculations were iterated by varying the initial
values of the complex refractive indices until both 1/ the adjusted value for the calculation of the
extended size distributions and 2/ the comparison between calculations and measurements of the
extinction and scattering coefficients agreed within errors. Results of these comparisons are
presented in the Supplementary material.
**3.2.    Constitution of the data set**
This section describes the choices of temporal and spatial coincidences adopted for the comparisons
between POLDER-3, AERONET and in situ data.
**3.2.1.  Coincidence with AERONET**
As described in previous evaluation studies of aerosol products derived from satellites (e.g., Bréon et
al., 2011), two approaches can be considered in order to compare coincident ground-based
photometer and satellite aerosol data. One option is to select only the closest (in time) photometer
measurement and the closest (in distance) satellite pixel from the photometer site. Another method



consist in performing averaging within a certain time window for photometer data, and a spatial
average of the satellite data within a given distance from the photometer site. Bréon et al. (2011) have
shown that these two approaches give very comparable results for POLDER-3 aerosol products over
oceans. In this study we adopted the second approach, considering the POLDER-3 aerosol products
from pixels within ± 0.5° around the AERONET sites. For AERONET AOD and AE, the averaging
temporal window was set to ±1 h around the time of the POLDER-3 passage. For AERONET $AOD_F$,
$AOD_C$, and shape retrieval, this temporal window produces an insufficient number of data, in particular
for springs and summers in the period 2005-2011 due to the temporal time shift of the POLDER-3
passage towards the afternoon. Instead, the averaging temporal window was extended to the whole
afternoon (that is, all data points later than 12:00 UTC) in order to allow for a significant dataset for
comparison.
**Table 1** reports the number of available observational days for POLDER-3 and AERONET aerosol
parameters at each station in the period March 2005-October 2013, as well as the number of
coincident days obtained between POLDER-3 and AERONET. The stations are ranged regarding the
number of coincident days obtained for AOD and AE, this number representing the upper limit of the
number of common POLDER-3/AERONET observations days available. Including all 17 stations,
18634 occurrences of comparable POLDER-3 and AERONET observations are available for AOD,
AE, $AOD_F$ and $AOD_C$, and 7923 occurrences for $AOD_{CS}$ and $AOD_{CNS}$, due to specific constraints on
geometric conditions in the POLDER-3 algorithm necessary to derive shape-related parameters (non
sphericity). Per site, the number of clear sky observational days for POLDER-3-derived AOD, AE,
$AOD_F$ and $AOD_C$ varies from 668 to 1392. Part of this variability also depends on the percent of sea
pixels in the 1° x 1° area around the sites, which is lower for coastal (e.g., Burjassot or Roma) than
insular stations (e.g., Alboran, Lampedusa or Gozo). Between 1 pixel in the case of inland stations of
Roma and Burjassot, and up to 29 pixels in the case of the small remote island of Alboran were
considered. Overall, the number of available AERONET observation days is important both for AOD
and AE (18223), and $AOD_F$ and $AOD_C$ (11228). The number of days with AERONET-derived $f_{NS}$ was





less significant (4976 data points), due to additional constrains in the inversion necessary to derive
this parameter.
The number of available AERONET observations per site varied from 158 to 2059 for AOD and AE,
and from 43 to 1333 for $AOD_F$ and $AOD_C$, mainly due to partial functioning of the instruments or
maintenance of the sites. At some stations, measurements started years after the beginning of
POLDER-3 mission (e.g., 2011 for Alboran, 2013 for Gozo). Finally, the number of POLDER-
3/AERONET coincident days available for analysis is 6421 for AOD and AE, 3855 for $AOD_F$ and
$AOD_C$, and 730 for the percentage of spherical coarse particles ($f_{NS}$).
**3.2.2.  Coincidence with airborne observations**
The comparison between POLDER-3 and airborne measurements was conducted for profile ascents
or descents of the ATR 42 close in time with POLDER-3 overpasses. Flight tracks and profiles
locations are shown in **Figure 3**, whereas additional details (dates, geographical coordinates, altitude
span and duration) are given in **Table 2**. Data from the PLASMA sunphotometer, operated only during
ADRIMED, were available only on 8 profiles (also indicated in Table 2) for which the minimum flight
altitude was as close as possible to the surface. The data set was limited to ATR 42 profiles extending
as much as possible over the column. To evaluate whether the aircraft profile sampled entirely or only
partially the aerosol layers, we compared the AOD measured by PLASMA to that obtained by
integrating the extinction profile of the CAPS-PMex instrument (not shown). By examining the
AERONET time series, we also excluded episodes when the AOD had significantly varied in time
between the POLDER-3 overpass and the aircraft profile. This mostly happened for cases when the
aerosol optical depth exceed 0.2 due to the transport of mineral dust (flights T-V22 and T-V23 during
TRAQA and V31-S3 and V42-S2 during ADRIMED). The profiles discarded for comparison with
POLDER-3 were used for the validation of the optical calculations presented in section 4 (not shown
in Table 2 nor Figure 3).
Prior to analysis, all in situ airborne data were synchronised and then averaged to 30 seconds to
reduce the noise due to the native resolution of the measurements (1 to 6 seconds). POLDER-3 data
were averaged over pixels within ±0.5° around the lowest altitude of each profile. In order to analyse





the aerosol vertical stratification, we examined the magnitude of the scattering coefficient $\sigma_{scatt}$ at 550
nm as a function of altitude and its spectral behaviour, represented by the scattering Angström
Exponent ($AE_{scatt}$) measured by the airborne nephelometer. As in previous similar studies (Pace et
al., 2006; Formenti et al., 2011; Di Biagio et al., 2015; 2016; Denjean et al., 2016), the aerosol layers
were classified in four categories (clear/background maritime, desert dust, pollution, and mixture),
following the criteria reported in **Table 3**. The mixture category, indicating mixing between desert dust
and pollution, as observed by Denjean et al. (2016), was further detailed to distinguish dust-dominated
layers ($AE_{scatt}$ between 0.5 and 0.75) and pollution-dominated layers ($AE_{scatt}$ between 0.75 and 1).
**3.3.    Statistical indicators**
The agreement between the POLDER-3, AERONET and airborne datasets was quantified by several
evaluation metrics, including the number of matchups (N), the linear correlation coefficient (R), the
slope (S) and intercept (I) of the linear regression, the root mean square error (RMS), and the bias
(B), representing their mean difference.

$$\text{RMS} = \sqrt{\tfrac{1}{n}\sum_{i=1}^{n}(y_i - x_i)^2} \tag{5}$$

$$B = \tfrac{1}{n}\sum_{i=1}^{n}(y_i - x_i) \tag{6}$$

where x and y are generic datasets, and n the number of pairs of compared values.
Additional metrics is provided by the "fraction of accurate retrievals" ($G_{frac}$) defined by Bréon et al.
(2011). This quantity is defined as

$$G_{frac} = \frac{\#obs(\Delta < EE)}{\#obs} \tag{7}$$

and quantifies the fraction of POLDER-3 data points for which the absolute difference ($\Delta$) between
reference and evaluated data is lower than the estimated error (EE).
In accordance to Bréon et al. (2011), EE was calculated as






$$EE = \pm (0.03 + 0.05 \times AOD) \qquad\qquad (8)$$

and applied to all the AOD advanced products. Because $G_{frac}$ is only appropriate for large datasets
whose number of data points exceeds 100 (Bréon et al., 2011), it was calculated only for comparisons
with AERONET data.
**4. Results**
**4.1.    Evaluation of the total aerosol optical depth**
**Figure 4** shows the results of comparison of the AOD retrieved by POLDER-3 between 2005 and
2013 with respect to the 6421 observations at the seventeen AERONET stations and those on the
vertical profiles of the ChArMEx campaigns (PLASMA sunphotometer and calculations from the in
situ size distributions).
The comparison with AERONET shows a good correlation (regression coefficient R = 0.88, $G_{frac}$ =
73%), with a statistically low dispersion and bias (RMS = 0.04, B = 0.003). Twenty-seven percent of
the observations do not meet the criteria of the $G_{frac}$ parameter. Cases outside the $G_{frac}$ boundary were
characterized by large standard deviations, either because the spatial distribution of AOD was
heterogeneous in the 1° x 1° area of the pixels surrounding the AERONET sites, or because it varied
significantly on the time window of ±1 hour around the POLDER-3 overpass. In our dataset, the
highest value of AOD measured by POLDER-3 was 1.4 (±0.1) during a desert dust transport event
over Lampedusa observed on April 25, 2011. This is the only event coincident with an AERONET
measurement (1.50 ±0.06) with POLDER-3 AOD >1.
Figure 4 also shows the comparison with the PLASMA observations and with the calculations initiated
by the measured airborne number size distributions.
On those, the AOD did not exceed 0.2, whereas AE ranged from 0.31 ±0.07 to 1.09 ±0.08, indicating
that these cases are representative of aerosols of different origins. The comparison was also very
satisfactory and confirmed the more extensive results from the comparison with AERONET-derived



AODs. POLDER-3 provides higher values of AOD for mineral dust (lowest AE values) compared to
those calculated from in situ aerosol measurements, which could reflect an underestimate of the
coarse mode distribution from the in situ aircraft measurements. On the other hand, POLDER-3 tends
to underestimate AOD with respect to PLASMA at low AE values, resulting in a negative bias of the
correlation (bias = -0.02). In both cases, the RMS remained low and below 0.05.
**4.2.    Evaluation of fine and coarse aerosol optical depth**
**4.2.1.  Comparison with AERONET observations**
**Figure 5** shows the comparison between POLDER-3 and AERONET for $AOD_F$ and $AOD_C$. $AOD_F$
remained below 0.25, smaller than $AOD_C$, which reached 0.8. The correlation coefficient for $AOD_C$ (R
= 0.81) is closer to the correlation coefficient for AOD (0.88) than that for $AOD_F$ (0.63). The agreement
between POLDER-3 and AERONET is confirmed by the $G_{frac}$ values of 74% for $AOD_C$ and 88% for
$AOD_F$, the low statistical bias (-0.007 for $AOD_F$ and 0.01 for $AOD_C$), and the moderate dispersion
(RMS values between 0.02 for $AOD_F$ and 0.04 for $AOD_C$). The weaker correlation and the dispersion
observed for $AOD_F$ can be attributed to the difficulty in retrieving low values of optical depth.
Additionally, Tanré et al. (2011) pointed out that differences could arise by the definitions of the cut-
off diameter ($D_{cut-off}$) used in the POLDER-3 and AERONET retrievals to estimate $AOD_F$. In the
AERONET retrievals, $AOD_F$ is calculated from the fine mode of the particle size distribution defined
for a value of $D_{cut-off}$ forced between 0.44 and 0.99 μm. In the POLDER-3 algorithm, $AOD_F$ is calculated
from the full particle size distribution of the retrieved fine mode, without cuf-off. However, because of
its use of polarisation, POLDER-3 is the most sensitive to particles smaller than 0.6-0.8 μm in
diameter (Tanré et al., 2011 and references therein).
In **Figure 6**, we explore the relevance of this difference in the comparison of $AOD_F$ and $AOD_C$ by
further separating days when AERONET $D_{cut-off}$ <1.0 μm and days when $D_{cut-off}$  ≥1.0 μm. The
threshold value of 1.0 μm corresponds to the $D_{eff}$ of all the fine modes in the POLDER-3 LUT. Cases
with $D_{cut-off}$ <1.0 μm were more numerous (2413 days), and showed a better agreement (Bias = -
0.003, $G_{frac}$ = 91%, RMS = 0.02, R = 0.60). Data corresponding to $D_{cut-off}$ ≥1.0 μm were less numerous
(1442 days). Whereas the correlation improved slightly (R = 0.69 versus R = 0.60), the dispersion



487 increased (bias = -0.01, RMS = 0.03) due to the appearance of points for which AERONET $AOD_F$

488 almost doubled that of POLDER-3. Colouring the data points by AE showed that the data points with

489 $D_{cut-off}$ below 1.0 μm mostly corresponded to aerosols with a weak-to-moderate spectral dependence

490 (low AE), whereas cases with $D_{cut-off}$ above 1.0 μm mostly (but not exclusively) corresponded to

491 aerosols with a moderate-to-strong spectral dependence (high AE).

492 The size cut-off definition also affects the comparison for $AOD_C$. For $D_{cut-off}$ <1.0 μm, $AOD_C$ values

493 were high and the correlation was significant. Conversely, $AOD_C$ remained low (≲0.2) when $D_{cut-off}$

494 ≥1.0 μm. This is consistent with the fact that the contribution of $AOD_C$ to AOD decreases as the $D_{cut-}$

495 $_{off}$ increases (**Figure S1** in the supplementary material). **Figure 6** shows that discriminating data on

496 the basis of $D_{cut-off}$ results in attributing $AOD_F$ and $AOD_C$ to different aerosol types.

497 **4.2.2. Comparison with airborne measurements**

498 To understand further the previous comparisons, POLDER-3 $AOD_F$ and $AOD_C$ were recalculated from

499 the measured number size distributions (Equation 4) by varying the lower limit of the size integration

500 between 0.4 and 1.0 μm in diameter with a step of 0.2. Results are shown in **Figure 7**. As expected,

501 the comparison for $AOD_F$ is very sensitive to the size range. The best agreement between the

502 retrieved and the calculated $AOD_F$ is obtained for $D_{cut-off}$ between 0.6 and 0.8 μm, both showing high

503 correlation coefficient R and low RMS. Conversely, the $AOD_C$ comparison is almost independent of

504 the value of $D_{cut-off}$ but more affected by the upper limit of the size range in Equation 4.

505 **4.3. Evaluation of the Ångström Exponent**

506 **Figure 8** shows the comparison of AE retrieved by POLDER-3 with values obtained by AERONET,

507 PLASMA and the optical calculations. The comparison with AERONET was restricted to days when

508 the POLDER-3 AOD exceed 0.1 (2031 out of the 6421 data points) to take into account only those

509 values with relative uncertainties within 50%. The comparison showed a significant spread and a

510 moderate correlation coefficient (R = 0.70). However, POLDER-3 tends to underestimate values of

511 AE larger than 1 with respect to AERONET, and overestimate values smaller than 0.5, yielding a

512 significant bias (–0.11). The values obtained by POLDER-3 compare well with the airborne





observations of PLASMA (R = 0.84), but less well to the optical calculations (R = 0.42). In both cases,
the bias is positive (0.1 with PLASMA and 0.2 with in situ AE). This fact, observed previously by
Goloub et al. (1999) and Tanré et al. (2011), can be explained by considering that the values of AE
are calculated from the retrieved AOD at 865 and 670 nm (Equation 1), which, in the ocean retrieval
algorithm of POLDER, is obtained by the fit of measured radiance. The current aerosol models in the
LUT (modal diameters and real part of the refractive index) provide AE values in the range –0.18 to
3.3. However, the extreme values are obtained only if the size distribution allowing to match the
observed radiances consists of a single mode of non-spherical coarse particle (modal diameter of 0.9
µm for AE = –0.18) or a single mode of fine spherical particles (modal diameter of 0.08 µm for AE =
3.3). **Figure 9** compares the scatterplots of AE and AOD obtained for the coincident POLDER-3 and
AERONET datasets. The tendency of POLDER-3 to underestimate AE shows up clearly by the
absence of values of AE larger than 2.5, which, conversely, are retrieved by AERONET. On the other
end of the spectrum, values down to –0.5 are found in the AERONET data set when POLDER-3
hardly retrieves negative values. Both POLDER-3 and AERONET show a trend with the largest AOD
values at lower AE values. However, high AOD values (>0.9) are found with POLDER but not
AERONET, and are all except one associated to relatively low AE (<1). Because the cloud screening
of AERONET is relatively robust thanks to triplet measurements (Smirnov et al., 2000), these outliers
may result from undetected cloud contamination in the POLDER algorithm.
**4.4.    Evaluation of aerosol sphericity**
When the geometrical conditions of observations are favourable, the coarse mode optical depth
(AOD$_C$) retrieved by POLDER-3 is quantified and apportioned into a spherical and a non-spherical
fraction (AOD$_{CS}$ and AOD$_{CNS}$, respectively). These products are potentially very useful in
discriminating the mineral dust contribution, dominated by non-spherical coarse particles (e.g.,
Dubovik et al., 2002; Chou et al., 2008), when marine aerosols can be considered as spherical at
relative humidities characteristics of coastal and open-sea sites (Sayer et al., 2012a; 2012b).
As a prerequisite, we investigated the comparison between POLDER-derived f$_{CNS}$ (percent fraction of
non-sphericity in the coarse mode AOD$_C$, that is, f$_{CNS}$ = AOD$_{CNS}$/(AOD$_{CNS}$+ AOD$_{CS}$) retrieved by



POLDER-3 and $f_{NS}$ (percent of non-sphericity of the total AOD) estimated by AERONET. In the
operational ocean algorithm, $f_{CNS}$ is a discrete value equal to 0, 0.25, 0.50, 0.75, and 1, but the
averaging process produces intermediate values when there is local variability between the pixels
around a given AERONET station
In general, the POLDER-3 $f_{CNS}$ and the AERONET $f_{NS}$ are poorly correlated. The correlation coefficient
R is 0.29 for the coincident data points of all the 17 stations (N = 730, **Table 1**). At individual stations,
notably the coastal and insular ones such as Lampedusa and Malaga, the correlation between
POLDER-3 $f_{CNS}$ and AERONET $f_{NS}$ is more significant (R = 0.73 for N = 54 and R = 0.59 for N = 53,
respectively). This is also seen when restricting the data set of Ersa and Lampedusa to the summers
of 2012 and 2013 (R = 0.55 at Ersa, N = 11; R = 0.70 at Lampedusa, N = 10).
The robustness of the comparison can be increased by further constraining the dataset to POLDER-
3 and AERONET AOD values larger than 0.10 and limiting the comparison to AERONET data for
which $AOD_C$ is at least 30% of the total AOD. By applying these thresholds (**Figure 10**), the correlation
between $f_{CNS}$ and $f_{NS}$ is R = 0.56 (N = 274 for the 17 stations). Overall, 80% of the POLDER-3 $f_{CNS}$
agrees within 25% with the AERONET values. The largest differences occur when AERONET
retrieves $f_{NS}$ values lower than 50%. In this case, only 40% of the POLDER-3 $f_{CNS}$ are in the ±25%
agreement interval with AERONET. Conversely, for AERONET $f_{NS}$ >50%, 88% of the POLDER-3 $f_{CNS}$
agree within ±25% with the AERONET estimate of $f_{NS}$. Finally, **Figure 11** shows that a relatively good
agreement is obtained by comparing broad classes 25% wide, providing consistency to the
classification of non-sphericity by POLDER-3.
**5.  Discussion**
**5.1.    Evaluation of uncertainties on the advanced POLDER-3 oceanic aerosol products**
In this paper we provide a first comprehensive evaluation of the advanced POLDER-3 aerosol
products over ocean by the latest operational algorithm, based on ground-based remote sensing
(AERONET) but also airborne remote sensing and in situ observations (TRAQA and ADRIMED
campaigns) over the western Mediterranean sea. **Table 4** summarizes it by presenting the absolute



errors (Δ) derived from the RMS (representing the precision) and the bias (B) as a measure of
accuracy. For consistency with previous similar analyses and as an acknowledgment of the large size
of the dataset, only the RMS and the bias of the linear regressions with the AERONET data have
been reported. The uncertainties in $AOD_{CS}$ and $AOD_{CNS}$ were calculated as the square-root of the
quadratic sum of the errors in $AOD_C$ and $f_{CNS}$.
Our estimate of ΔAOD indicates that, for the western Mediterranean basin, the accuracy and the
precision of the POLDER-3 are better than those derived by the error analysis of Tanré et al. (2011),
also reported in Table 4, based on a global comparison with AERONET of the POLDER-1 instrument.
It is noteworthy that the POLDER-1 retrieval algorithm was using a single mode spherical particle size
distribution (Goloub et al., 1999) instead of the current two modes allowing an aspherical component.
Furthermore, from our regional evaluation of the whole latest collection 3 of the POLDER-3 data set,
$G_{frac}$ value for AOD (73%) is much better than that reported by Bréon et al. (2011) ($G_{frac}$= 45%), based
on previous collection of POLDER-3 retrievals at a global scale.

**5.2.    Evaluation of the fine and coarse AOD**

Table 4 reports the uncertainties in $AOD_F$ and $AOD_C$ based on estimates RMS and bias. It is
interesting to notice that the precision in $AOD_C$ is apparently lower than in $AOD_F$ (higher RMS), despite
the correlation being far better for the former than for the latter. We have shown that the direct
comparison between POLDER-3 and AERONET should take into account the differences in the
definition of the fine size fraction in the respective retrieval algorithms. The AERONET $AOD_F$ is
recalculated from the fine mode of the volume size distribution retrieved from the measured total
radiance, and defined as the mode below an upper limit diameter ($D_{cut-off}$) varying between 0.88 and
1.98 µm. Conversely, our comparison with airborne measurements indicates that the $AOD_F$ retrieved
by POLDER-3 corresponds to a fine mode extending to values of $D_{cut-off}$ between 0.6 and 0.8 µm. This
is expected as POLDER-3 uses polarised radiances, highly sensitive to fine particles, in agreement
with previous regional validations of POLDER $AOD_F$ over land (Kacenelenbogen et al., 2006; Fan et
al., 2008; Wang et al., 2015). The comparison with in situ data shows that the POLDER-3 $AOD_C$ is





less sensitive to the $D_{cut-off}$ value (**Figure 7**), but mostly to the extent of the coarse mode towards the
largest particles.

**5.3.     Regional aerosol distribution**

The ability of POLDER-3 in representing the spatial distribution of aerosols in the Mediterranean
region is demonstrated in Figure 12 showing the retrieved products averaged over the operating
period. These regional maps highlights a north-south gradient for AOD and $AOD_{CNS}$, with, on average,
the highest values in the southernmost part of the western Mediterranean region, especially over
south Ionian Sea off Libya, as previously reported by former satellites AOD products (e.g., Moulin et
al., 1998; Antoine and Nobileau, 2006). The distribution of POLDER-3 AE indicates high values along
the European coasts (especially over the Adriatic Sea), and low along the North African coasts
indicative of the dominance of desert dust in the South and anthropogenic aerosol in the North of the
basin. $AOD_F$ and $AOD_{CS}$ maps show moderate spatial variability over the basin, associated to
averaged values ($AOD_F$ of 0.033, $AOD_{CS}$ of 0.021) 2 to 3 times lower than those retrieved by
POLDER-3 for $AOD_{CNS}$ (0.065). Despite these low spatial patterns, it is noticeable that $AOD_F$ values
tend to increase in the Eastern part of our region of study, suggesting the complexity of various aerosol
types influences over the Mediterranean Sea.
The detailed investigation of the aerosol climatology and regional distribution of the POLDER-3
derived aerosol optical depth load of the fine and coarse mode aerosol, including spherical and non-
spherical components, over the western Mediterranean Sea, as a support to the ongoing research in
the area, will be presented in a companion paper

**6.  Conclusive remarks**

The western Mediterranean aerosol is a complex mixture with a significant temporal and spatial
variability at small scales (Pace et al., 2005; 2006; Di Iorio et al., 2009; Mallet et al., 2016 and
references therein), and significant impact on present and future regional climate (Nabat et al., 2014;
2015a; 2015b; 2016). High-resolved long-time series of spaceborne observations of aerosol optical
depth on different size classes and for differing particle shapes, such as provided by POLDER-3, are

essential in exploring those evolutions, directly, but also indirectly, as a term of comparison for climate and transport models (Nabat et al., 2014). In the past, quantitative remote sensing of the AOD has proven most useful in establishing decadal climatology of the transport of mineral dust over the basin, highlighting its seasonal variability, geographic distribution and sources, link to large-scale atmospheric dynamics (Dulac et al., 1992; Moulin et al., 1997a; 1997b; 1998; Antoine and Nobileau, 2006; Papadimas et al., 2008).

The quality of the observations is surely key to those surveys, and has motivated the comparative analysis of the advanced POLDER-3 oceanic aerosol products during the whole period of operation (March 2005 to October 2013) presented in this paper, with regards to co-located and coincident ground-based measurements by AERONET, and airborne vertical profiles of aerosol optical depth and size distribution during the TRAQA and ADRIMED campaign of the ChArMEx project.

The results presented in this paper confirm previous validations (Goloub et al., 1999; Kacenelenbogen et al., 2006; Fan et al., 2008; Bréon et al., 2011; Tanré et al., 2011), and provide a first evaluation of the uncertainties on the fine and coarse fractions of the aerosol optical depth, and the partitioning of the coarse mode AOD into its spherical and non-spherical components. They allow moving forward in the classification of the Mediterranean aerosols, and in particular in the investigation of the anthropogenic fraction, which is relevant to climate change. As a matter of fact, our results indicate that the fine-fraction AOD at 865 nm can be constrained to the aerosol accumulation mode below 0.6-0.8 µm in diameter. This suggests that the $AOD_F$ measured by POLDER-3 could be used for predicting the submicron column concentrations for air quality studies, and for evaluating the radiative effect of fine aerosols.

**Data availability**

POLDER-3 data extraction was performed with the program PARASOLASCII (http://www-loa.univ-lille1.fr/~ducos/public/parasolascii/). This version is made available from the AERIS Data and Service Center (http://www.icare.univ-lille1.fr/parasol). The AERONET version 2.0 aerosol products at the level 2.0 quality (cloud screened and quality assured with up-to-date calibration) were obtained from the official website at http://aeronet.gsfc.nasa.gov/. Single particle Mie scattering calculations were



performed with the Mie_single.pro routine under IDL available at
http://eodg.atm.ox.ac.uk/MIE/mie_single.html.

**Competing interests**

The authors declare that they have no conflict of interest.

**Special issue statement**

This article is part of the special issue "CHemistry and AeRosols Mediterranean Experiments
(ChArMEx) (ACP/AMT inter-journal SI)". It is not associated with a conference.

**Acknowledgements**

This work is part of the ChArMEx project supported by CNRS-INSU, ADEME, Météo-France and CEA
in the framework of the multidisciplinary program MISTRALS (Mediterranean Integrated Studies aT
Regional And Local Scales; http://mistrals-home.org/). It has also been supported by the French
National Research Agency (ANR) through the ADRIMED program (contract ANR-11-BS56-0006) and
by the French National Program of Spatial Teledetection (PNTS, http://www.insu.cnrs.fr/pnts, project
n°PNTS-2015-03). L. Mbemba Kabuiku was granted by the French Environment and Energy
Management Agency (ADEME) and National Center of Space Studies (CNES). Airborne data was
obtained using the ATR-42 atmospheric research aircraft managed by Safire, which is a joint facility
of the French national center for scientific research (CNRS), Météo-France and CNES. The AERIS
national data infrastructure provided access to the POLDER-3 data used in this study. Teams from
AERONET and its French component PHOTONS are acknowledged for calibrating the sun
photometer network and producing long-term time series of quality assured aerosol product time
series used in this study. We thank the AERONET principal investigators L. A. Arboledas (Alboràn),
S. Basart and J. M. Baldasano (Barcelona), B. N. Holben (Blida), J. A. Martinez Lozano (Burjassot),
M. Mallet (Ersa and Montesoro Bastia), P. Goloub (Ersa), J. Piazzola (Frioul and Porquerolles), R.
Ellul (Gozo), D. Meloni (Lampedusa), F. J. Olmo Reyes (Malaga), S. Pignatti (Messina), J. R. Moreta
Gonzalez (Palma de Mallorca), G. P. Gobbi (Rome), Z. Ameur (Tizi Ouzou), S. Despiau (Toulon) and
D. Antoine (Villefranche-sur-Mer) and their staff for establishing and maintaining the 17 sites used in



this investigation. C. Di Biagio (LISA) and C. Denjean (CNRM) are acknowledged for help with data
analysis. G. Siour (LISA) is acknowledged for help with figure production.



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



**Figure captions**

**Figure 1.** Map of the location of the 17 AERONET ground-based stations considered in this work.

**Figure 2**. Iterative data inversion procedure to retrieve from airborne observations the aerosol optical depth (AOD, $AOD_F$ and $AOD_C$) and Angstrom exponent (AE) as measured by POLDER-3. Green boxes indicate the input values from airborne measurements (size distribution, scattering and extinction coefficients) and the initial values of the complex refractive indices estimated from published literature. The iterative steps of the procedure are indicated in the blue boxes. The results of optical calculations (corrected size distribution, scattering and extinction coefficients) are in the orange boxes.

**Figure 3.** Flight tracks of the ATR 42 aircraft (coloured lines) during the TRAQA (left) and ADRIMED (right) campaigns. Only flights relevant to this study are presented. The location of the vertical profiles coincidental, at their lowermost altitude, with a POLDER-3 overpass is shown by a black star.

**Figure 4.** Scatter plots of daily AOD retrieved by POLDER-3 at 865 nm with respect to: (top panel) coincident and co-located values from the 17 ground-based AERONET sites at 870 nm; (middle panel) airborne PLASMA sunphotometer operated at 865 nm during ADRIMED; (bottom panel) results of the optical calculations at 865 nm according to Figure 2 from airborne measurements during TRAQA and ADRIMED. The solid line is the bisector. The dashed lines represent the limits indicated by the $G_{frac}$ parameter. The characteristics of the linear correlation (number of points Nb, correlation coefficient, $G_{frac}$, RMS and bias) are also reported.

**Figure 5.** Scatter plots of daily $AOD_F$ and $AOD_C$ retrieved by POLDER-3 at 865 nm as a function of coincident AERONET values at 870 nm for the 17 sites in the western Mediterranean. The solid line is the bisector. The dashed lines represent the limits indicated by the $G_{frac}$ parameter. The characteristics of the linear correlation (number of points Nb, correlation coefficient, $G_{frac}$, RMS and bias) are also reported.

**Figure 6.** Same as figure 5 for cases corresponding to AERONET retrievals yielding a separation of the fine and coarse modes of the volume distribution at $D_{cut-off} < 1.0$ μm (left) and days with AERONET $D_{cut-off} \geq 1.0$ μm (right).

**Figure 7.** Scatter plots of $AOD_F$ (left) and $AOD_C$ (right) retrieved by POLDER-3 at 865 nm and compared to values obtained by optical calculations from airborne measurements of the particle size number distribution. Panels, from top to bottom, represent the results of the calculations when varying the cut-off diameter between 0.4 and 1.0 μm. Characteristics of the linear correlation are also reported (number of points Nb, correlation coefficient R, RMS and bias). Error bars of in situ measurements were calculated from the optical calculation and the instrumental uncertainties. The solid line is the bisector.

**Figure 8.** Scatter plots of the Angström Exponent (AE) retrieved by POLDER-3 between 865 and 670 nm with respect to coincident and collocated values from: (top) the 17 ground-based AERONET sites between 870 and 675 nm; (middle) airborne PLASMA sun photometer operated at 870 and 675 nm during ADRIMED; (bottom) optical calculations at 865 and 670 nm from particle size number distributions measured in situ during TRAQA and ADRIMED. Only AERONET values corresponding to POLDER-3 AOD >0.1 are considered because of large uncertainties in AE at low AOD. To facilitate the reading, the standard deviations of the AERONET values are not represented. Characteristics of the linear correlations are also reported (number of points Nb, correlation coefficient R, RMS and bias).

**Figure 9.** Scatter plot of AE versus AOD retrieved by POLDER-3 (left) and AERONET (right) on coincidental days (N=6421) for the 17 stations of Western Mediterranean Sea. Mean and standard deviations (in brackets) of AOD obtained by classifying the air masses into pollution (blue, AE ≥ 1.5), mixed (green, 0.5 < AE < 1.5) and desert dust (orange, AE ≤ 0.5) according to Pace et al. (2006) are shown.

**Figure 10.** Scatterplot of the fraction of coarse mode optical depth due to non-spherical particles ($f_{cns}$) retrieved by POLDER-3 and that of total optical depth ($f_{ns}$) estimated by AERONET. Values are



expressed in percent. Only AERONET data points for which the measured AOD exceeded 0.10 and
the $AOD_C$ represented more than 30% of the total AOD are represented. The solid line is the bisector.
Dashed lines represent the interval of ± 25% of agreement between POLDER-3 $f_{CNS}$ and AERONET
$f_{NS}$.
**Figure 11.** Mean and standard deviations of coarse mode optical depth due to non-spherical particles
measured by POLDER-3 ($f_{cns}$, blue) and that of total optical depth estimated by AERONET ($f_{ns}$, red)
classified into four classes: spherical ($f_{cns}$ ≤25%); predominant spherical (25%< $f_{cns}$ ≤50%),
predominant non-spherical (50%< $f_{cns}$ ≤75%); non-spherical (75%< $f_{cns}$ ≤100%). Values are
expressed in percent. Only AERONET data points for which the AOD >0.10 and $AOD_C$/AOD >0.30
are represented. The black triangles represent the number of points in each classes (the dashed
curves is represented for increased readability).
**Figure 12.** Regional maps of average AOD (top left), AE, (top right), $AOD_F$ (middle left), $AOD_C$ (middle
right), $AOD_{CNS}$ (top left), and $AOD_{CS}$ (bottom right) retrieved by POLDER-3 over the period March
2005-October 2013. Mean and standard deviations are also shown.





**Table captions**

**Table 1.** List of AERONET stations available in the western Mediterranean region with at least one ocean POLDER pixel ($Nb_{POL}$) within 0.5° around the station, together with the number of POLDER-3 vs. AERONET observations (and coincident days in brackets) for the different aerosol products from March 2005 to October 2013.

**Table 2.** List of vertical profiles made by the ATR 42 aircraft during the TRAQA and ADRIMED campaigns in coincidence with the passage of POLDER-3. For each profile is indicated: the flight number, the name of the profile, the date, the time period of the profile, the area covered by the flight, the geographical coordinates, the minimum and maximum altitude of the flight and then, the hour of POLDER-3 overpass in UTC.

**Table 3.** Criteria of classification of aerosol layers encountered on the vertical profiles of TRAQA and ADRIMED, based on nephelometer measurements of the scattering coefficient ($\sigma_{scatt}$) at 550 nm and on its spectral dependence ($AE_{scatt}$) between 450 and 700 nm.

**Table 4.** Summary of evaluated uncertainties on POLDER-3 advanced aerosol products AOD, AE, $AOD_F$, $AOD_C$, and $f_{CNS}$. N/A stands for not attributed.





**Table 1.** List of AERONET stations available in the western Mediterranean region with at least one ocean POLDER pixel (Nb$_{POL}$) within 0.5° around the station, together with the number of POLDER-3 vs. AERONET observations (and coincident days in brackets) for the different aerosol products from March 2005 to October 2013.

| AERONET station | Latitude, Longitude | Altitude (m) | AERONET period | Nb$_{POL}$ | AOD and AE | AOD$_F$ and AOD$_C$ | f$_{CNS}$ and f$_{NS}$ |
|---|---|---|---|---|---|---|---|
| | | | | | POLDER/AERONET (coincidences) | | |
| Barcelona | 41°23'N, 02°07'E | 125 | 04/03/2005 - 10/10/2013 | 13 | 1171/2059 (827) | 1171/1333 (514) | 485/623 (116) |
| Villefranche-sur-Mer | 43°41'N, 07°19'E | 130 | 17/02/2005 - 21/08/2013 | 9 | 1097/1589 (641) | 1097/999 (359) | 470/452 (77) |
| Toulon | 43°08'N, 06°00'E | 50 | 04/03/2005 - 04/12/2010 | 9 | 1114/1503 (630) | 1114/962 (343) | 429/393 (67) |
| Ersa | 43°00'N, 09°21'E | 80 | 09/06/2008 - 11/10/2013 | 17 | 1178/1252 (541) | 1178/676 (281) | 504/240 (37) |
| Malaga | 36°42'N, 04°28'W | 40 | 23/02/2009 - 23/09/2013 | 10 | 1193/1359 (539) | 1193/1036 (419) | 465/377 (53) |
| Lampedusa | 35°31'N, 12°37'E | 45 | 06/03/2005 - 11/10/2013 | 28 | 1301/1177 (513) | 1301/663 (307) | 604/285 (54) |
| Messina | 38°11'N, 15°34'E | 15 | 01/05/2005 - 23/20/2012 | 9 | 1119/1340 (507) | 1119/739 (281) | 538/399 (63) |
| Roma Tor Vergata | 41°50'N, 12°38'E | 130 | 10/03/2005 - 11/10/2013 | 1 | 725/1954 (486) | 725/1199 (280) | 297/683 (66) |
| Blida | 36°30'N, 02°52'E | 230 | 06/03/2005 - 19/02/2012 | 7 | 989/1357 (475) | 989/813 (280) | 427/484 (85) |
| Burjassot | 39°30'N, 00°25'W | 30 | 16/04/2007 - 24/04/2013 | 1 | 668/1506 (372) | 668/1045 (277) | 249/480 (54) |
| Palma de Mallorca | 39°33'N, 02°37'E | 10 | 03/08/2011 - 10/10/2013 | 11 | 1136/524 (214) | 1136/395 (155) | 504/162 (19) |
| Porquerolles | 43°00'N, 06°09'E | 22 | 10/05/2007 - 17/07/2013 | 11 | 1106/537 (195) | 1106/260 (95) | 431/82 (9) |
| Frioul | 43°15'N, 05°17'E | 40 | 07/07/2010 - 11/10/2013 | 8 | 1037/481 (162) | 1037/324 (118) | 373/91 (10) |
| Gozo | 36°02'N, 14°15'E | 32 | 25/02/2013 - 11/10/2013 | 24 | 1320/210 (102) | 1320/162 (67) | 633/90 (9) |
| Montesoro Bastia | 42°40'N, 09°26'E | 49 | 26/07/2012 - 23/07/2013 | 14 | 1161/240 (76) | 1161/43 (7) | 506/12 (1) |
| Alboran | 35°56'N, 03°02'E | 15 | 29/06/2011 - 23/01/2012 | 29 | 1392/158 (73) | 1392/103 (46) | 609/47 (7) |
| Tizi Ouzou | 36°41'N, 04°03'E | 133 | 11/04/2012 - 11/10/2013 | 5 | 927/238 (68) | 927/98 (26) | 399/76 (3) |
| TOTAL | - | - | - | - | 18634/18223 (6421) | 18634/11228 (3855) | 7923/4976 (730) |





**Table 2.** List of vertical profiles made by the ATR 42 during the TRAQA and ADRIMED campaigns in coincidence with the passage of POLDER-3. For each profile is indicated: the flight number, the name of the profile, the date, the time period of the profile, the area covered by the flight, the geographical coordinates, the minimum and maximum altitude of the flight and then, the hour of POLDER-3 overpass in UTC.

| Campaign | Flight ID | Profile ID | Date | Time (UTC) | Area | Geographical span Beginning | Geographical span end | Altitude span (m asl) | POLDER-3 overpass (UTC) | PLASMA |
|---|---|---|---|---|---|---|---|---|---|---|
| TRAQA | T-V21 | T-V21-S1 | 27/06/2012 | 10h31–10h52 | Corse | 42°59'N–7°43'E | 42°59'N–7°41'E | 122–3534 | 14h19 | ---- |
| | T-V24 | T-V24-S1 | 03/07/2012 | 15h39–16h08 | N-East Barcelona | 42°14'N–3°31'E | 42°8'N–3°29'E | 77–3832 | 15h03 | ---- |
| | T-V25 | T-V25-S1 | 04/07/2012 | 08h32–09h04 | South of France–Lion Gulf | 41°28'N–6°0'E | 41°31'N–6°0'E | 100–4444 | 14h05 | ---- |
| | T-V26 | T-V26-S1 | | 16h08–16h41 | | 42°45'N–4°13'E | 42°46'N–4°13'E | 128–4684 | | ---- |
| | T-V27 | T-V27-S1 | 06/07/2012 | 09h01–09h26 | South of France | 42°41'N–5°19'E | 42°39'N-5°14'E | 115–4723 | 13h47 | ---- |
| | | T-V27-S3 | | 09h26–11h00 | | 42°39'N-5°15'E | 42°42'N–5°19'E | 76–3782 | | ---- |
| | T-V28 | T-V28-S2 | | 16h20–16h42 | | 42°19'N–7°35'E | 42°44'N–6°22'E | 60–3784 | | ---- |
| ADRIMED | A-V28 | A-V28-S2 | 14/06/2013 | 10h19–10h44 | East Corse–Sardinia | 41°38'N–7°14'E | 42°4'N–6°46'E | 69–3860 | 14h56 | Yes |
| | A-V29 | A-V29-S1 | 16/06/2013 | 08h19–08h32 | Baleares–Sardinia | 39°15'N–9°3'E | 39°40'N–8°59'E | 6–3877 | 14h37 | Yes |
| | | A-V29-S4 | | 09h46–10h15 | | 39°34'N-4°29'E | 39°39'N-4°29'E | 52–4521 | | Yes |
| | A-V30 | A-V30-S1 | 16/06/2013 | 11h59-12h10 | Baleares–Sardinia | 39°52'N-4°13'E | 39°32'N-3°48'E | 93–3240 | 14:37 | Yes |
| | A-V31 | A-V31-S4 | 17/06/2013 | 09h41–09h54 | Baleares–Sardinia | 40°11'N–3°59'E | 39°52'N–4°13'E | 95–2899 | 15h18 | Yes |
| | A-V32 | A-V32-S1 | | 11h46–12h05 | | 39°52'N-4°13'E | 39°56'N-4°36'E | 93–4519 | | Yes |
| | | A-V32-S4 | | 13h30–13h44 | | 39°32'N–9°10'E | 39°16'N–9°2'E | 10–3548 | | Yes |
| | A-V33 | A-V33-S2 | 19/06/2013 | 12h47–13h17 | Corse–Sardinia | 43°01'N–9°23'E | 43°1'N–9°20'E | 73–4502 | 15h00 | Yes |
| | | A-V33-S4 | | 14h46–14h59 | | 39°15'N–9°24'E | 39°15'N–9°4'E | 5–3224 | | ---- |
| | A-V38 | A-V38-S2 | 28/06/2013 | 12h25–13h30 | Sardinia–Lampedusa | 35°30'N–12°38'E | 35°30'N–12°37'E | 12–5427 | 14h26 | ---- |
| | A-V44 | A-V44-S1 | 04/07/2013 | 12h22–12h33 | Gulf of Genoa–Corse–Sardinia | 43°02'N–9°15'E | 43°2'N–9°19'E | 59–3513 | 15h11 | ---- |
| | | A-V44-S2 | | 14h35–14h51 | | 43°35'N–9°7'E | 39°15'N–9°4'E | 4–3499 | | ---- |



**Table 3.** Criteria of classification of aerosol layers encountered on the vertical profiles of TRAQA and
ADRIMED, based on nephelometer measurements of the scattering coefficient ($\sigma_{scatt}$) at 550 nm and
on its spectral dependence ($AE_{scatt}$) between 450 and 700 nm.

| Aerosol type | $AE_{scatt}$(450-700 nm) | $\sigma_{scatt}$ (550 nm) |
|---|---|---|
| Clean background / maritime | – | < 5 or 10 Mm$^{-1}$ |
| Desert dust | < 0.5 | > 10 Mm$^{-1}$ |
| Pollution | > 1 | |
| Mixed (dust-dominated) | 0.5 – 0.75 | > 10 Mm$^{-1}$ |
| Mixed (pollution-dominated) | 0.75 – 1 | |





Table 4. Summary of evaluated uncertainties on POLDER-3 advanced products AOD, AE, AOD$_F$,
AOD$_C$, and f$_{CNS}$, and comparison to previous evaluations. N/A stands for not attributed.


| Products | Uncertainties | |
|---|---|---|
| | **This work** | **Previous work** |
| **AOD** | $\Delta AOD = \pm(0.003 + 0.04 \times AOD)$ | $\Delta AOD = \pm(0.05 \times AOD + 0.05)^\$$ |
| **AE** | $\Delta AE = \pm(0.11 + 0.44 \times AE)$ | $\Delta AE = 0.3–0.5^\$$ |
| **AOD$_F$** | $\Delta AOD_F = \pm(0.007 + 0.02 \times AOD_F)$ | N/A |
| **AOD$_F$ (D$_{cut\text{-}off}$ < 1 μm)** | $\Delta AOD_F = \pm(0.003 + 0.02 \times AOD_F)$ | N/A |
| **AOD$_C$** | $\Delta AOD_C = \pm(0.01 + 0.04 \times AOD_C)$ | N/A |
| **f$_{NCS}$** | $\Delta f_{CNS} = \pm25\%$ | N/A |
| **AOD$_{CS}$** | $\Delta AOD_{CS} = AOD_{CS} \times [(0.04 + 0.01/AOD_{CNS})^2 + ((1-\Delta f_{CNS})/(1-f_{CNS}))^2]^{1/2}$ | N/A |
| **AOD$_{CNS}$** | $\Delta AOD_{CNS} = AOD_{CNS} \times [(0.04 + 0.01/AOD_{CNS})^2 + (\Delta f_{CNS}/f_{CNS})^2]^{1/2}$ | N/A |

$^\$$ Tanré et al., (2011) and references therein





**Figure 1.** Map of the location of the 17 AERONET ground-based stations considered in this work.

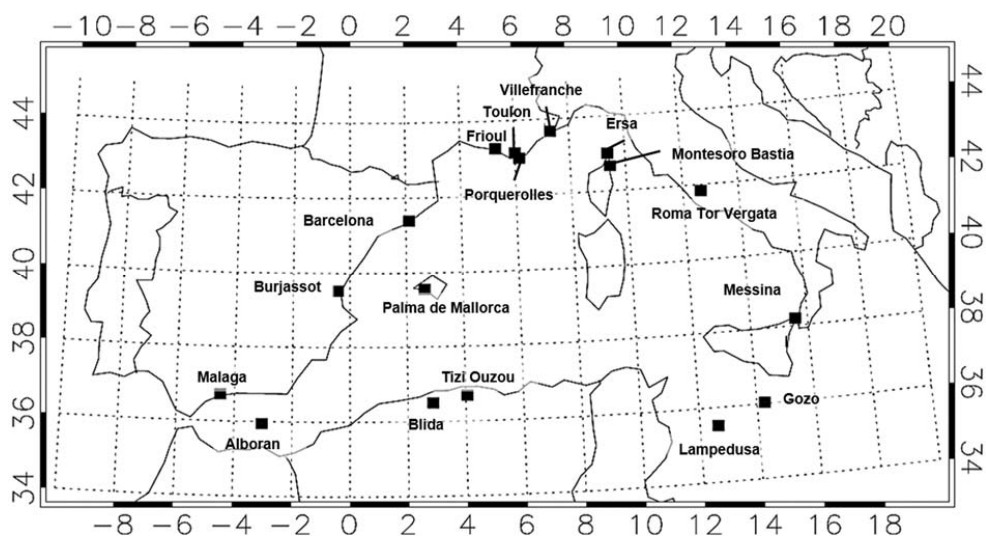







**Figure 2**. Iterative data inversion procedure to retrieve the aerosol optical depth (AOD, $AOD_F$ and $AOD_C$) and Angstrom exponent (AE) measured by POLDER-3 from airborne observations. Green boxes indicate the input values from airborne measurements (size distribution, scattering and extinction coefficients) and the initial values of the complex refractive indexes estimated from published literature. The iterative steps of the procedure are indicated in the blue boxes. The results of optical calculations (corrected size distribution, scattering and extinction coefficients) are in the orange boxes.

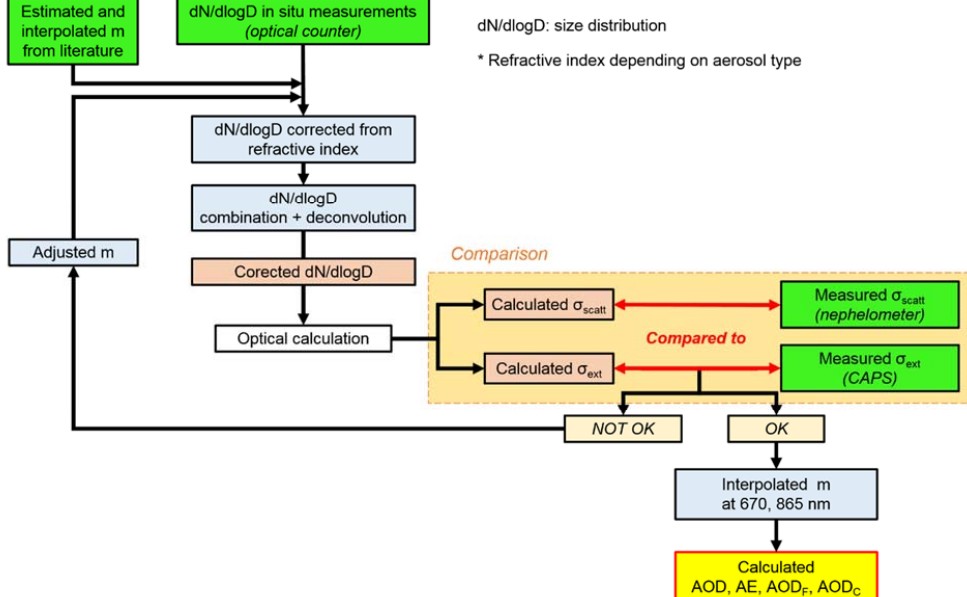



**Figure 3.** Flight tracks of the ATR 42 aircraft (coloured lines) during the TRAQA and ADRIMED
campaigns. Only flights relevant to this study are presented. The location of the profiles coincidental,
at their lowermost altitude, with a POLDER-3 overpass is shown by a circle. During the TRAQA
campaigns, 7 profiles were retained for comparison on 6 flights. During the ADRIMED campaign, 12
profiles occurring during 9 flights were retained. In this second case, symbols are not always visible
as overlapping.

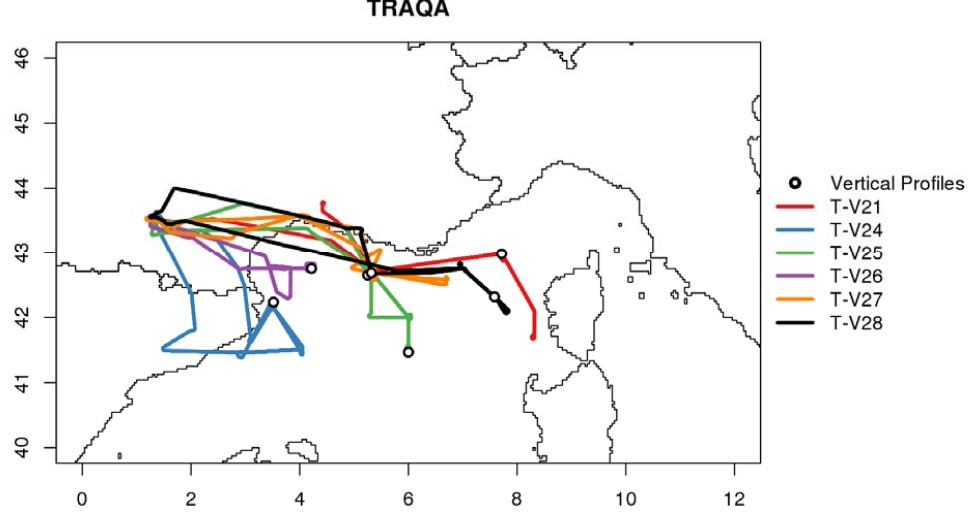


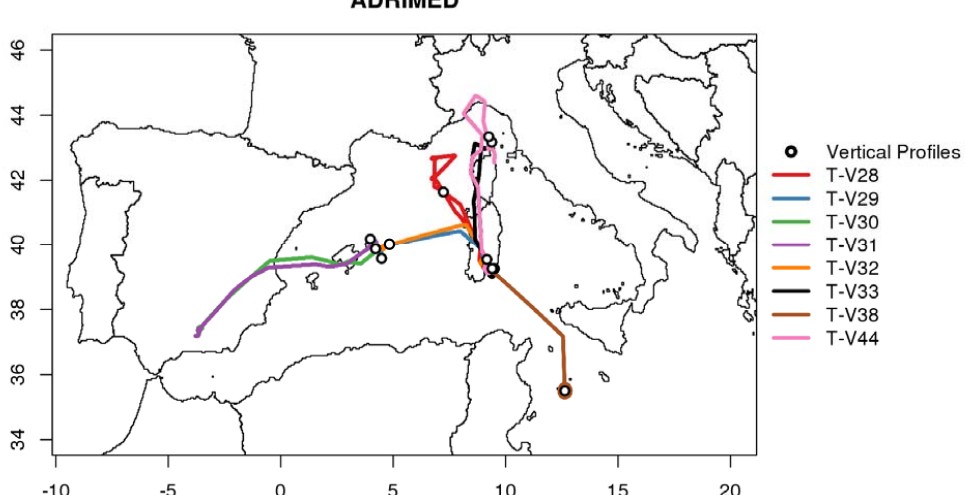






**Figure 4.** Scatterplots of daily AOD retrieved by POLDER-3 at 865 nm with respect to: (top panel) coincident and co-located values from the 17 ground-based AERONET sites at 870 nm; (middle panel) airborne PLASMA sunphotometer operated at 865 nm during ADRIMED; (bottom panel) results of the optical calculations at 865 nm according to Figure 1 from airborne measurements during TRAQA and ADRIMED. The solid line is the bisector. The dashed lines represent the limits indicated by the $G_{frac}$ parameter. The characteristics of the linear correlation (number of points, correlation coefficient, $G_{frac}$, RMS and bias) are also reported.

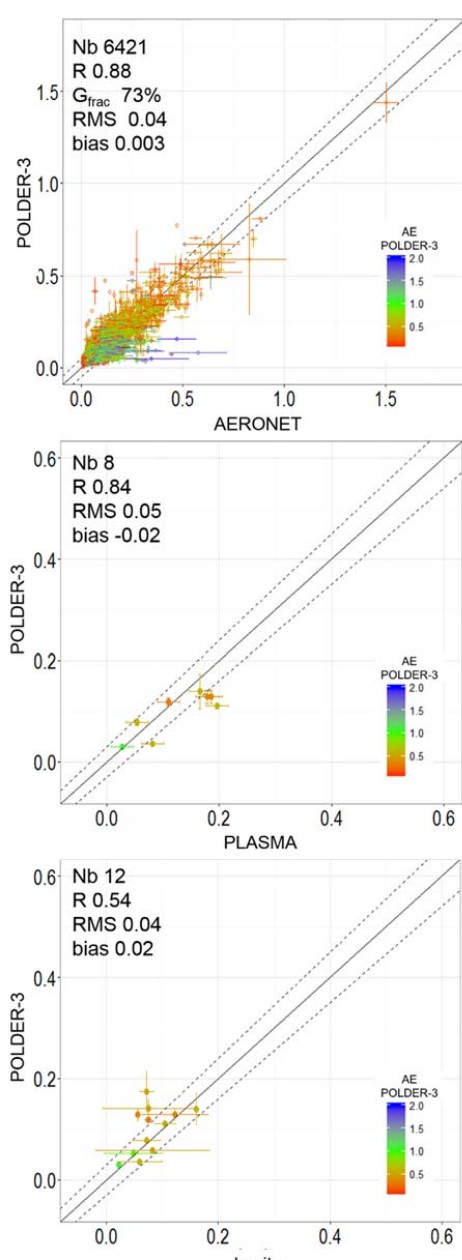



**Figure 5.** Scatter plots of daily AOD$_F$ and AOD$_C$ retrieved by POLDER-3 at 865 nm as a function of
coincident AERONET values at 870 nm for the 17 sites of Western Mediterranean Sea. The solid line
is the bisector. The dashed lines represent the limits indicated by the G$_{frac}$ parameter. The
characteristics of the linear correlation (number of points, correlation coefficient, G$_{frac}$, RMS and bias)
are also reported.

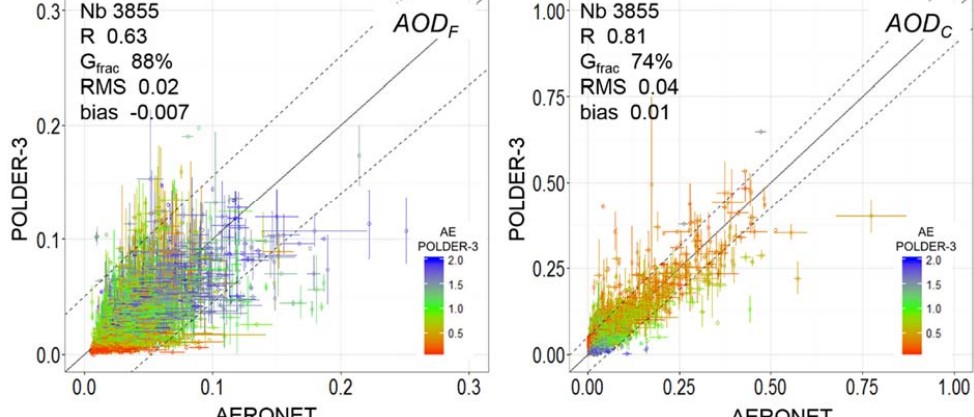






**Figure 6.** Scatter plots of daily AOD$_F$ (top) and AOD$_C$ (bottom) retrieved by POLDER-3 at 865 nm as function of coincident AERONET values at 870 nm at the 17 sites of Western Mediterranean Sea for cases corresponding to AERONET retrievals yielding a separation of the fine and coarse modes of the volume distribution at D$_{cut-off}$ < 1.0 μm (left) and days with AERONET D$_{cut-off}$ ≥ 1.0 μm (right). The solid line is the bisector. The dashed lines represent the limits indicated by the G$_{frac}$ parameter. The characteristics of the linear correlation (number of points, correlation coefficient R, G$_{frac}$, RMS and bias) are also reported.

**(a) AOD$_F$**

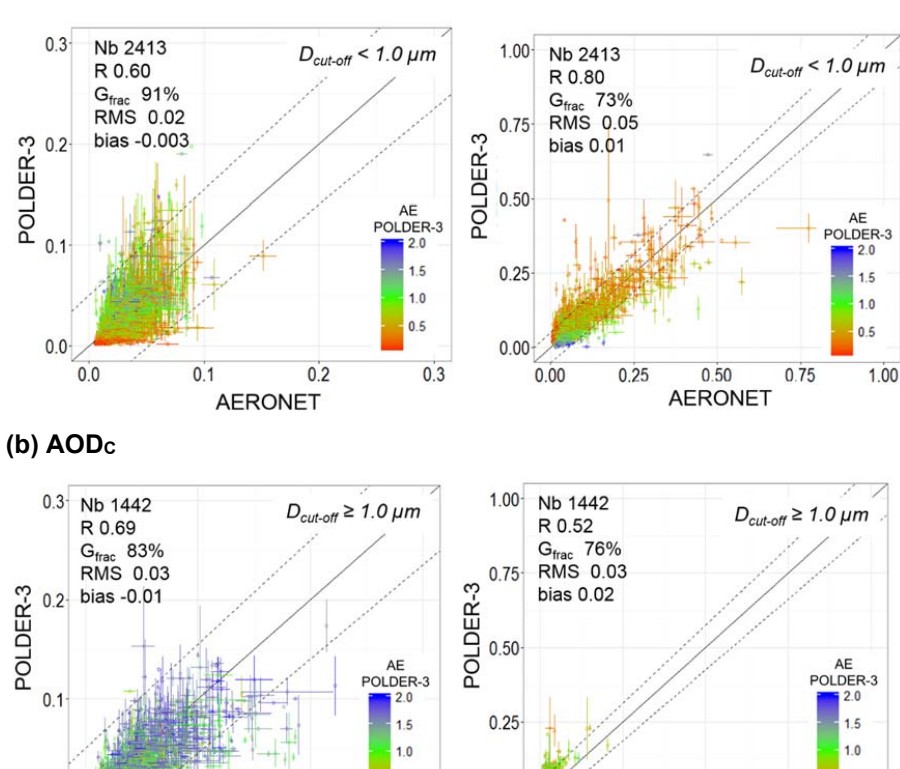

**(b) AOD$_C$**





**Figure 7.** Scatter plots of $AOD_F$ (left) and $AOD_C$ (right) retrieved by POLDER-3 at 865 nm and compared to values obtained by optical calculations from airborne measurements of the number size distribution. Panels, from top to bottom, represent the results of the calculations when varying the cut-off diameter between 0.4 and 1.0 μm. Characteristics of the linear correlation are also reported (number of points, correlation coefficient R, RMS and bias). Error bars of in situ measurements were calculated from the optical calculation and the instrumental uncertainties. The solid line is the bisector.

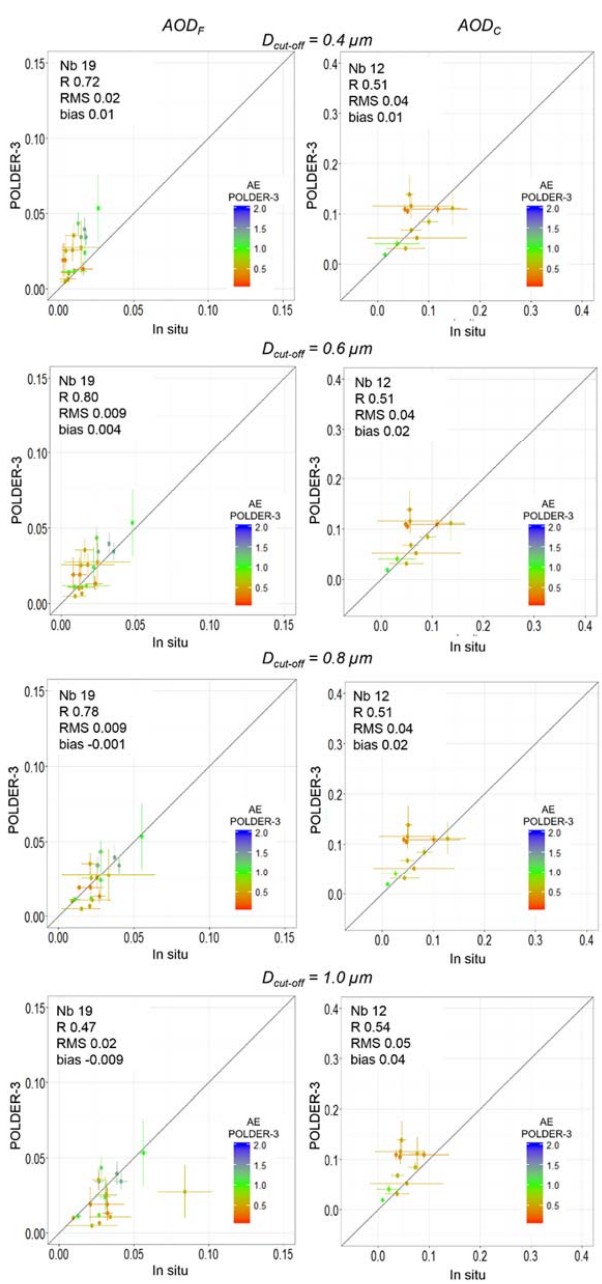





**Figure 8.** Scatter plots of the Angström Exponent (AE) retrieved by POLDER-3 between 865 and 670
nm with respect to coincident and collocated values from (top) the 17 ground-based AERONET sites
between 870 and 675 nm; (middle) airborne PLASMA sunphotometer operated at 870 and 675 nm
during ADRIMED; (bottom) optical calculations at 865 and 670 nm from number size distributions
measured in situ during TRAQA and ADRIMED. Only AERONET values corresponding to POLDER-
3 AOD larger than 0.1 are considered. To facilitate the reading, the standard deviations of the
AERONET values are not represented. Characteristics of the linear correlations are also reported
(number of points, correlation coefficient R, *RMS* and bias).

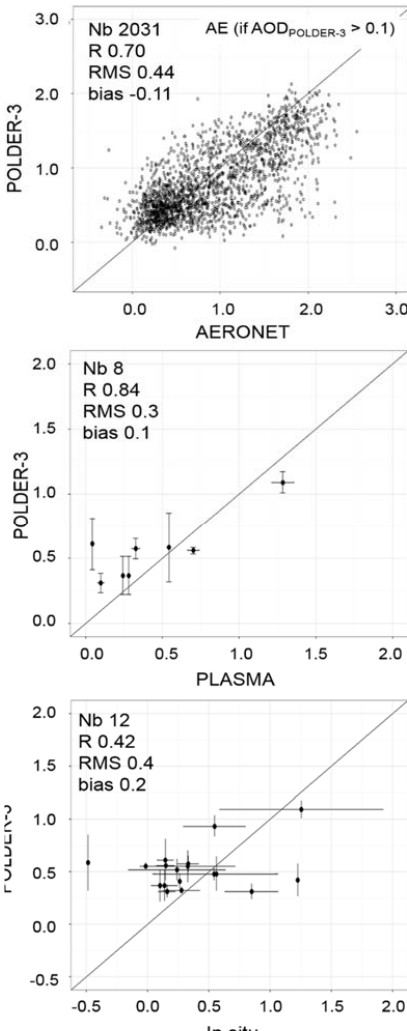






**Figure 9.** Scatter plot of AE versus AOD retrieved by POLDER-3 (left) and AERONET (right) on coincidental days (N=6421) for the 17 stations of Western Mediterranean Sea. Mean and standard deviations (in brackets) of AOD obtained by classifying the air masses into pollution (blue, AE ≥ 1.5), mixed (green, 0.5 < AE < 1.5) and desert dust (orange, AE ≤ 0.5) according to Pace et al. (2006) are shown.

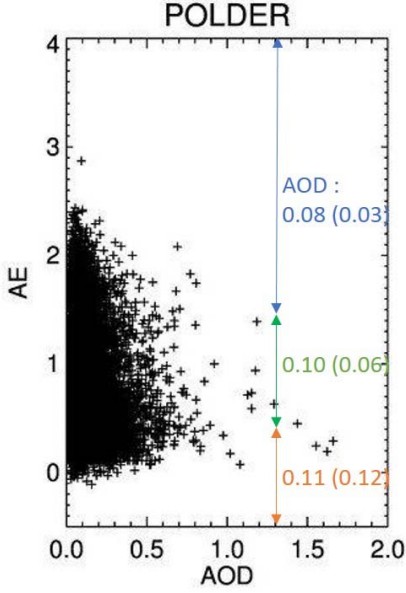
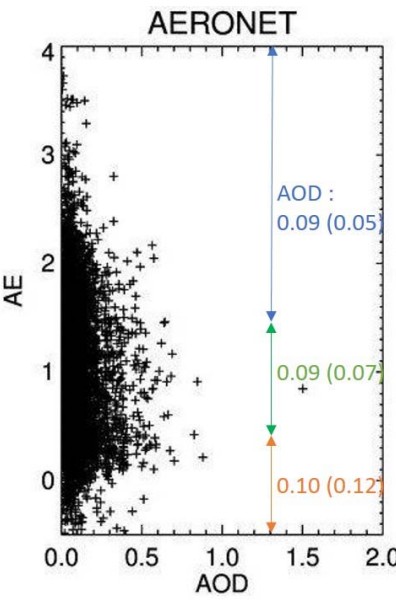



**Figure 10.** Scatterplot of the fraction of coarse mode optical depth due to non-spherical particles (f$_{cns}$) retrieved by POLDER-3 and that of total optical depth (f$_{ns}$) estimated by AERONET. Values are expressed in percent. Only AERONET data points for which the measured AOD exceeded 0.10 and the AOD$_C$ represented more than 30% of the total AOD are represented. The solid line is the bisector. Dashed lines represent the interval of ± 25% of agreement between POLDER-3 f$_{CNS}$ and AERONET f$_{NS}$.



**Figure 11.** Mean and standard deviations of coarse mode optical depth due to non-spherical particles
measured by POLDER-3 ($f_{cns}$, blue) and that of total optical depth estimated by AERONET ($f_{ns}$, red)
classified into four classes: spherical ($f_{cns}$ ≤25%); predominant spherical (25%< $f_{cns}$ ≤50%),
predominant non-spherical (50%< $f_{cns}$ ≤75%); non-spherical (75%< $f_{cns}$ ≤100%). Values are
expressed in percent. Only AERONET data points for which the AOD >0.10 and $AOD_C$/AOD >0.30
are represented. The black triangles represent the number of points in each classes (the dashed
curves is represented for increased readability).

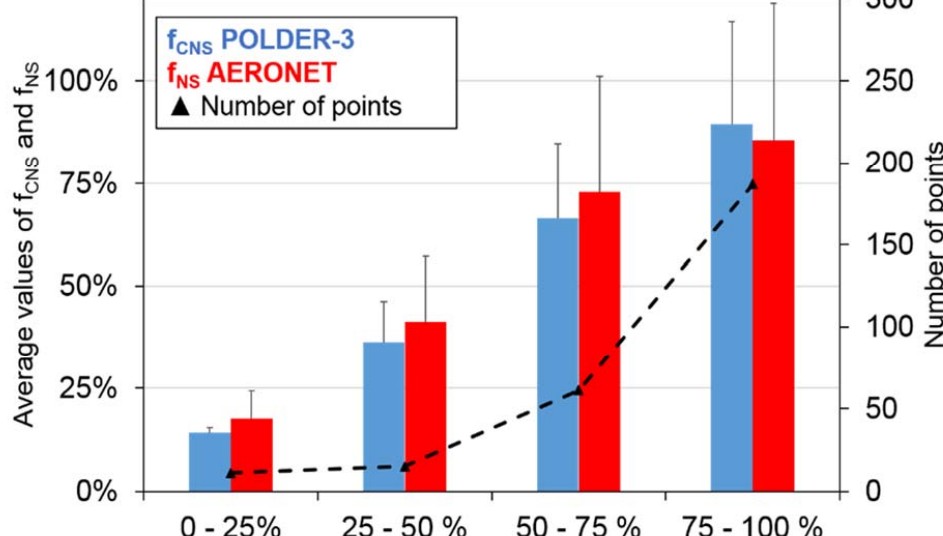






**Figure 12.** Regional maps for AOD, AE, AOD$_F$ (top panel from left to right), AOD$_C$, AOD$_{CNS}$ and AOD$_{CS}$
(bottom panel from left to right) retrieved by POLDER-3 for the period March 2005-October 2013.
Mean and standard deviations are also shown.
