# Peer review of "Aerosol optical properties derived from POLDER-3/PARASOL (2005-2013) over the western Mediterranean Sea: I. Quality assessment with AERONET and in situ airborne observations"

_Atmospheric Measurement Techniques, 2018_

## Referee Comment (RC1) · Anonymous Referee #1 · 13 Sep 2018

The authors present a 2-part analysis of POLDER-3/PARASOL oceanic aerosol retrievals against ground-based AERONET validation (in the Mediterranean), as well as a comparison of different sub-orbital (in-situ) data taken in the region. For the former, the authors present compelling evidence of POLDER-3 sensitivity to aerosol size, fine/coarse mode discrimination, AOD, and non-sphericity (to some extent). For the latter, the authors compare results from different optical-particle counters, providing a nice summary of retrieved complex refractive index for different aerosol types. The authors have clearly performed a thorough literature review, and this work should be

published after minor revisions.

General Comments:

I would strongly encourage the authors to convert AOD, fine-mode AOD, and coarse-mode AOD to 550 nm rather than 865 nm. Many other retrieval algorithms provide AOD information (such as MODIS DT) at this wavelength (or at least near it), and solar irradiance is much higher (meaning absolute attenuation will be larger) at 550 nm. Fine-mode AOD is typically very small at 865 nm, which will result in a lower RMSE and correlation as compared to the coarse mode (which you see in Figures 4 and 5). I expect that your fine-mode AOD range will more than double by extrapolating to 550 nm, and I expect your RMSE to increase substantially too. Although the lack of absorption is probably not an issue because your retrieved fine-mode AODs are so low (and desert dust is non-absorbing in the red and NIR), you may see a low bias in AODf at 550 nm because the effects of absorption can lead to non-linear errors in retrieved AOD.

As POLDER's sensitivity to sphericity is probably dependent on total aerosol loading, might it make sense to report non-spherical AOD rather than non-spherical AOD fraction?

I might be a bit biased towards the POLDER-3/AERONET analysis, but I think the paper might flow better if all of the in-situ analysis were moved to the supplemental (or into its own paper). It really seems like an add-on to the POLDER-3/AERONET work.

Specific Comments:

Line 255: Should read "can be calculated as".

Line 365: Is this increased temporal window only for AODF and AODC, or for all measurements?

Line 452: I think this should read "retrieved" not "measured", as POLDER does not measure AOD.

Line 580-582: At the risk of sounding like a broken record, I believe that this can be explained by your use of 865nm AOD rather than 550 nm AOD. Table 4: The uncertainties here do not make sense to me [maybe I am just missing something?]: 1. Your RMSE is substantially larger than the absolute term in your AOD uncertainty (which you have as an extremely low 0.003 [should this be 0.03?]) 2. AE uncertainty should be a function of AOD or just a flat envelope. The higher the AOD, the greater confidence you should have in particle properties. 3. Non-spherical AOD uncertainty makes a lot more sense than fNCS uncertainty, as you can account for inherent bias at low AOD.

Figure 2-3: I would move this to supplemental, but up to you.

Figure 4: I would remove the bottom to panels, as you have too few data to provide anything of value from airborne. Maybe then merge Figure 4 with 5?

Figure 6: There appear to be a couple of issues with this figure: 1. Should the caption read "volume distribution at Dcut-off < 1.0 $\mu$m (left) and days with AERONET Dcut-off $\geq$ 1.0 $\mu$m (right)" or "volume distribution at Dcut-off < 1.0 $\mu$m (Top) and days with AERONET Dcut-off $\geq$ 1.0 $\mu$m (Bottom)" 2. Figure 6 reads as though retrieved fine-mode AOD is the top plot, and coarse-mode AOD is the bottom plot. a. I assume that this is a mistake, and that the fine-mode retrievals are on the left, and the coarse-mode retrievals are on the right side. b. This should also be clarified in the caption.

Figure 7: I would move this to the supplemental as well.

Figure 8: I would change this to being contingent on AERONET AOD > 0.1, but this is just my preference. I would also remove the airborne data, as there are too few data. Maybe instead you could have 3 plots of AE, with different AOD requirements for each ( >0.05, >0.1, >0.2)? This would help demonstrate the dependence of AE errors on AOD.

Figure 9: This figure might make more sense as a color-density plot.

Figure 10: Would it make sense to change this to AODNS vs AODCNS?

[Figure]

---

## Referee Comment (RC2) · Anonymous Referee #2 · 1 Oct 2018

General comments:

The topic of the study is very important: an analysis of the quality of POLDER satellite measurements of aerosol properties over the Mediterranean. This analysis and error information can then be used by other researchers in the CHARMEX project.

The paper constitutes a very comprehensive study, and gives a clear overview of the Aeronet and aircraft measurements, together with their error sources. The thorough discussion of measurement methods and their errors and characteristics, including the

supplementary material, is welcomed and is an excellent example for other similar studies.

The paper is well written. The methods are well described, with extensive referencing. However, some figures could be clarified (see comments below).

In the introduction the title should be explained. The reader may wonder what the topic of part 2 will be. This should be clarified, e.g. at the end of the discussion. The fact that the interesting Figure 12 is only given at the end of the paper is probably a cliff-hanger to paper 2 ?

There is no information on trends in aerosols over the West-Mediterranean from POLDER and Aeronet data. That is a pity – is 8 years too short? Or will the trends be described elsewhere?

Specific comments:

1. Please say in the introduction why there is no attention given to the spatial distribution of aerosols in the West-Mediterranean area. The text given on lines 608-611 should be given in the introduction as well.

2. l. 131 ff: All symbols, like $m$, $D$, etc., should be in italics (slant font). This does not hold for acronyms, like AOD.

3. Header Table 1: Nbpol is an unclear quantity; please define.

4. Table 4: AOD, AE, etc. are acronyms and not symbols, so they should be in upright font.

5. Figure 2: What do the green boxes mean?

6. Caption Figure 4: What does daily AOD mean in the case of a polar orbiting satellite at 13:30? The individual data points of POLDER averaged over the 1x1 deg2 box?

7. Figure 4: Why is Nb used instead of N for the number of points?

8. In Figure 4 there are too many points to clearly see the correlation. Could you make a logarithmic AOD scale, to better zoom-in on the small values?

9. Caption Figure 5: Note that the definition of fine and coarse modes is probably not the same for POLDER and Aeronet.

10. Figure 6: I find this figure difficult to understand. D_cut-off is the threshold value itself, so it should be D > D_cutoff and D < D_cutoff. Is here D_cutoff itself a variable quantity ? I also do not understand the difference between the left and right figures.

11. Please always give the physical quantity in the axis label, so e.g. in Fig. 5, 6, and 7 AOD should be given in the label.

12. Figure 9: Please indicate the three AE ranges with horizontal boundary lines.

13. Caption Fig. 10: ns > NS. Please say that f_NS also is a fraction in terms of total optical depth.

14. Caption Fig. 12: Please use capitals for CNS, NS, ….

15. Caption Fig. 12: the AOD > AOD, classes > class, curves > curve

16. Concluding remarks: Could a recommendation be added on how to determine the cut-off diameter between fine and coarse aerosols?

17. Suppl. Table S1: what is the imaginary part of the refractive index?

18. Suppl. l. 24: at which wavelength does this refractive index value hold?

19. Suppl. l. 74: change > changes

20. Suppl. Table S3: please use a better alignment of words and numbers to avoid ugly breaks.

21. Suppl. L. 168: particle > particles

Textual corrections / suggestions:

l. 33: particles

l. 48: anthropic > anthropogenic

l. 76-77: Here is probably meant: multi-spectral imagery instruments

l. 86: operational ocean retrieval algorithm

l. 121: solar radiance > Earth radiance

l. 131: nul > zero

l. 160: due to rounding errors

l. 185: By clear sky > For clear sky

l. 255: car > can

l. 326: the atmosphere

l. 403: exceeded

l. 611: …...paper.

l. 612: Conclusive > Concluding

---

## Author Comment (AC1) · 15 Nov 2018

**Answer to reviews for ms amt-2018-25 - Formenti et al., Aerosol optical properties derived from POLDER-3/PARASOL (2005-2013) over the western Mediterranean Sea: I. Quality assessment with AERONET and in situ airborne observations**

We thank Referee #1 for evaluating the manuscript and providing us with feedback on its scientific content. Detailed responses are presented in the body of text here below in blue.

**Anonymous Referee #1**

The authors present a 2-part analysis of POLDER-3/PARASOL oceanic aerosol retrievals against ground-based AERONET validation (in the Mediterranean), as well as a comparison of different sub-orbital (in-situ) data taken in the region. For the former, the authors present compelling evidence of POLDER-3 sensitivity to aerosol size, fine/coarse mode discrimination, AOD, and non-sphericity (to some extent). For the latter, the authors compare results from different optical-particle counters, providing a nice summary of retrieved complex refractive index for different aerosol types. The authors have clearly performed a thorough literature review, and this work should be published after minor revisions.

General Comments:

I would strongly encourage the authors to convert AOD, fine-mode AOD, and coarse mode AOD to 550 nm rather than 865 nm. Many other retrieval algorithms provide AOD information (such as MODIS DT) at this wavelength (or at least near it), and solar irradiance is much higher (meaning absolute attenuation will be larger) at 550 nm. Fine-mode AOD is typically very small at 865 nm, which will result in a lower RMSE and correlation as compared to the coarse mode (which you see in Figures 4 and 5). I expect that your fine-mode AOD range will more than double by extrapolating to 550 nm, and I expect your RMSE to increase substantially too. Although the lack of absorption is probably not an issue because your retrieved fine-mode AODs are so low (and desert dust is non-absorbing in the red and NIR), you may see a low bias in AODf at 550 nm because the effects of absorption can lead to non-linear errors in retrieved AOD. As POLDER's sensitivity to sphericity is probably dependent on total aerosol loading, might it make sense to report non-spherical AOD rather than non-spherical AOD fraction?

We understand and appreciate the comments by Referee #1. It is true that the choice of wavelength is of importance: 865 nm results in small values of fine-mode AOD (compared to 550 nm), but it is a question of accuracy. The objective of the paper is the validation of the POLDER-3 retrievals at the wavelengths where the instrument made the measurements and the oceanic algorithm is applied. These are 865 and 670 nm, but not 550 nm. Converting all data to 550 nm would result in inducing an additional bias due to the limitations in the retrieval of the Angstrom exponent (AE). This is why, as a first step, it is of first importance to evaluate the retrieval at the instrument/algorithm wavelength. However, we will certainly consider the conversion for the second part of this paper, which will address the analysis of the AOD products for the investigation of the aerosol spatial distribution and temporal variability in the western Mediterranean. A sentence on this issue has been added in section 5.2.

I might be a bit biased towards the POLDER-3/AERONET analysis, but I think the paper might flow better if all of the in-situ analysis were moved to the supplemental (or into its own paper). It really seems like an add-on to the POLDER-3/AERONET work.

We considered in deep detail this suggestion by Referee #1. Our feeling, and the motivation behind the analysis, is that the comparison with the in situ data provides with additional information which augments the results obtained by the comparison with AERONET. In particular, they allow investigating the sensitivity to size of POLDER-3 retrievals. In this respect, we would prefer keeping them with the main text. This would imply keeping Figure 3, bottom panels of Figure 4, and Figure 7. The

alternative suggestion by Referee #1 is that the in situ-POLDER comparison could make the object of a paper *per se*. Again, we felt that the complementary of AERONET and in situ is the added value of the paper. We prefer to gather the available information in a single paper, the approach is rather original and we believe it gives more value to our study. On the contrary, it seems to us that there is not enough supplementary material for writing a solid additional paper.

Specific Comments:

Line 255: Should read "can be calculated as".

Done

Line 365: Is this increased temporal window only for AODF and AODC, or for all measurements?

This was done only for $AOD_F$ and $AOD_C$. To clarify the sentence has been changed from "Instead, the averaging temporal window was extended to the whole afternoon (that is, all data points later than 12:00 UTC) in order to allow for a significant dataset for comparison" to "For these two variables, the averaging temporal window was extended to the whole afternoon (that is, all data points later than 12:00 UTC) in order to allow for a significant dataset for comparison".

Line 452: I think this should read "retrieved" not "measured", as POLDER does not measure AOD.

Correct - Done

Line 580-582: At the risk of sounding like a broken record, I believe that this can be explained by your use of 865nm AOD rather than 550 nm AOD.

A sentence has been added.

Table 4: The uncertainties here do not make sense to me [maybe I am just missing something?]: 1. Your RMSE is substantially larger than the absolute term in your AOD uncertainty (which you have as an extremely low 0.003 [should this be 0.03?]) The 0.003 corresponds to Bias value reported in Figure 4.

2. AE uncertainty should be a function of AOD or just a flat envelope. The higher the AOD, the greater confidence you should have in particle properties.

The AE uncertainty is expressed as a function of AE from RMS and Bias values obtained in Figure 8, as done for AOD from Figure 4. The error is larger for larger AE which corresponds to lower AOD values.

3. Non-spherical AOD uncertainty makes a lot more sense than $f_{NCS}$ uncertainty, as you can account for inherent bias at low AOD.

The POLDER-3 oceanic algorithm retrieves $f_{NCS}$, which can only assume fixed values (0, 25, 50, 75 and 100%), without interpolation, and not the $AOD_{CNS}$. In this methodological paper it is therefore logic to evaluate this quantity and not the AOD products. We agree with the reviewer that $f_{NCS}$ poses problems when the AOD is low, that is why the product is provided only for AOD > 0.1.

Figure 2-3: I would move this to supplemental, but up to you.

We agree in moving Figure 2 but would prefer keeping Figure 3 in the main text as it is the parallel to Figure 1.

Figure 4: I would remove the bottom to panels, as you have too few data to provide anything of value from airborne. Maybe then merge Figure 4 with 5?

Again, we believe in the added values of the comparison to the in situ data, albeit based on a limited number of data points. The current representation is simple and easy to read. We would like to keep it as it is.

Figure 6: There appear to be a couple of issues with this figure: 1. Should the caption read "volume distribution at Dcut-off < 1.0 $\mu$m (left) and days with AERONET Dcut-off $\geq$ 1.0 $\mu$m (right)" or "volume distribution at Dcut-off < 1.0 $\mu$m (Top) and days with AERONET Dcut-off $\geq$ 1.0 $\mu$m (Bottom)" 2. Figure 6 reads as though retrieved finemode AOD is the top plot, and coarse-mode AOD is the bottom plot. a. I assume that this is a mistake, and that the fine-mode retrievals are on the left, and the coarse-mode retrievals are on the right side. b. This should also be clarified in the caption.

The reviewer is correct: the fine-mode AOD is the top plot and the coarse-mode AOD is the bottom plot. This is now corrected. The caption should read "the caption read "volume distribution at Dcut-off < 1.0 μm (left) and days with AERONET Dcut- off ≥ 1.0 μm (right)"

Figure 7: I would move this to the supplemental as well.

See previous comments. We would like to leave this in the main text.

Figure 8: I would change this to being contingent on AERONET AOD > 0.1, but this is just my preference. I would also remove the airborne data, as there are too few data. Maybe instead you could have 3 plots of AE, with different AOD requirements for each ( >0.05, >0.1, >0.2)? This would help demonstrate the dependence of AE errors on AOD.

The scope of this figure is not to show how the error on AE changes with increasing AOD bur rather how it compares to the AERONET retrieval when the right screening of AOD by POLDER-3 is done. Again the airborne data are few but illustrative.

Figure 10: Would it make sense to change this to AODNS vs AODCNS?

As we explained previously, the POLDER-3 oceanic algorithm retrieves $f_{NCS}$, which can only assume fixed values (0, 25, 50, 75 and 100%), without interpolation, and not the $AOD_{CNS}$. To clarify this, the text in lines 153-161 has been reworded. Because of that, we would like to keep the figure as it is. We have therefore added Figure 10 to show the scatterplot comparison between the POLDER-3 AODCNS and the AERONET AODNS. The 2 quantities are strongly correlated (R=0.87) but the POLDER-3 $AOD_{CNS}$ is lower than the AERONET $AOD_{NS}$, as expected. Explaining text has been added in Section 4.4. Former Figure 10 is now included as Figure 9.b.

---

## Author Comment (AC2) · 15 Nov 2018

**Answer to reviews for ms amt-2018-25 - Formenti et al., Aerosol optical properties derived from POLDER-3/PARASOL (2005-2013) over the western Mediterranean Sea: I. Quality assessment with AERONET and in situ airborne observations**

We thank Referee #2 for evaluating the manuscript and providing us with feedback on its scientific content. Detailed responses are presented in the body of text here below in blue.

**Anonymous Referee #2**

General comments:

The topic of the study is very important: an analysis of the quality of POLDER satellite measurements of aerosol properties over the Mediterranean. This analysis and error information can then be used by other researchers in the CHARMEX project.

The paper constitutes a very comprehensive study, and gives a clear overview of the Aeronet and aircraft measurements, together with their error sources. The thorough discussion of measurement methods and their errors and characteristics, including the supplementary material, is welcomed and is an excellent example for other similar studies.

The paper is well written. The methods are well described, with extensive referencing. However, some figures could be clarified (see comments below).

In the introduction the title should be explained. The reader may wonder what the topic of part 2 will be. This should be clarified, e.g. at the end of the discussion. The fact that the interesting Figure 12 is only given at the end of the paper is probably a cliff-hanger to paper 2 ? There is no information on trends in aerosols over the West-Mediterranean from POLDER and Aeronet data. That is a pity – is 8 years too short? Or will the trends be described elsewhere?

We have now added a sentence in the introduction and modified the sentence at the end of the discussion to clarify that the topic of part 2 would be the analysis of spatial distribution and temporal variability, including trends, provided by the analysis of POLDER-3 retrievals over the western Mediterranean.

Specific comments:

1. Please say in the introduction why there is no attention given to the spatial distribution of aerosols in the West-Mediterranean area. The text given on lines 608-611 should be given in the introduction as well.

This is now done

2. l. 131 ff: All symbols, like m, D, etc., should be in italics (slant font). This does not hold for acronyms, like AOD.

This is now done

3. Header Table 1: Nbpol is an unclear quantity; please define.

To increase readability, the caption was changed as "Table 1. List of AERONET stations available in the western Mediterranean region retained for this study. The number of ocean POLDER pixel within 0.5° from the position of the station is indicated ($N_{PIXEL}$). The number of observations by POLDER-3 and AERONET between March 2005 to October 2013, and the number of coincident days (within brackets) are also reported." Nbpol was changed into $N_{PIXEL}$.

4. Table 4: AOD, AE, etc. are acronyms and not symbols, so they should be in upright font.

This is now done.

5. Figure 2: What do the green boxes mean?

As indicated in the caption, green boxes indicate the input values from airborne measurements (size distribution, scattering and extinction coefficients) and the initial values of the complex refractive indexes estimated from published literature.

6. Caption Figure 4: What does daily AOD mean in the case of a polar orbiting satellite at 13:30? The individual data points of POLDER averaged over the 1x1 deg2 box?

Yes, daily indicates the average of individual data points of PODLER averaged over the 1x1 degree box every day.

7. Figure 4: Why is Nb used instead of N for the number of points?

This is now corrected.

9. Caption Figure 5: Note that the definition of fine and coarse modes is probably not the same for POLDER and Aeronet.

A sentence has been added to the Figure caption.

10. Figure 6: I find this figure difficult to understand. D_cut-off is the threshold value itself, so it should be D > D_cutoff and D < D_cutoff. Is here D_cutoff itself a variable quantity ? I also do not understand the difference between the left and right figures.

There was a problem with the order of panels in Figure 6 (see answer to Reviewer #1) which is now corrected. In Lines 180-182 we clarify this point but modifying the sentence as "The fine and coarse modes of the retrieved volume size distribution are defined as the modes below and above a threshold diameter ($D_{cut-off}$) corresponding to the minimum of the size distribution. The $D_{cut-off}$ value is not fixed but can vary between 0.44 and 0.99 µm". This sentence is also added to the Figure caption.

11. Please always give the physical quantity in the axis label, so e.g. in Fig. 5, 6, and 7 AOD should be given in the label.

This is now done.

12. Figure 9: Please indicate the three AE ranges with horizontal boundary lines.

This is now done.

13. Caption Fig. 10: ns > NS. Please say that f_NS also is a fraction in terms of total optical depth.

Done

14. Caption Fig. 12: Please use capitals for CNS, NS, …..

Done

15. Caption Fig. 12: the AOD > AOD, classes > class, curves > curve

Done

16. Concluding remarks: Could a recommendation be added on how to determine the cut-off diameter between fine and coarse aerosols?

A sentence has been added

17. Suppl. Table S1: what is the imaginary part of the refractive index?

The imaginary part of the refractive index is zero in the ocean retrieval algorithm. This is now added in the caption of Table S1

18. Suppl. l. 24: at which wavelength does this refractive index value hold?

The wavelength value has been added

19. Suppl. l. 74: change > changes

Done

20. Suppl. Table S3: please use a better alignment of words and numbers to avoid ugly breaks.

Done

21. Suppl. L. 168: particle > particles

Done

All the textual corrections / suggestions have been accepted

---

## Author Response (AR2)

**Answer to reviews for ms amt-2018-25 - Formenti et al., Aerosol optical properties derived from POLDER-3/PARASOL (2005-2013) over the western Mediterranean Sea: I. Quality assessment with AERONET and in situ airborne observations**

Dear editor,

Thank you for the acceptance of our revised manuscript. We have performed all the technical corrections requested.

The reference to where the quality flag index is defined has been added in the "Data availability" section as follows:

"POLDER-3 data extraction was performed with the program PARASOLASCII (http://www-loa.univ-lille1.fr/~ducos/public/parasolascii/). This version is made available from the AERIS Data and Service Center (http://www.icare.univ-lille1.fr/parasol). Technical details are described at http://www.icare.univ-lille1.fr/projects_data/parasol/docs/Parasol_Level-2_format_latest.pdf. The definition of the flag index is detailed at page 18 (parameter: quality of the fit)".

Best wishes,

Paola